# DIFFUSION MODELS IMPROVE ADVERSARIAL ROBUSTNESS BY COMPRESSING IMAGE SPACE

## ABSTRACT

Recent work suggests that diffusion models significantly enhance empirical adversarial robustness. While several intuitive explanations have been proposed, the precise mechanisms remain unclear. In this work, we systematically investigate how diffusion models improve adversarial robustness. First, we observe that diffusion models intriguingly increase, rather than decrease, the $\ell_p$ distance to clean samples—challenging the notion that purification denoises inputs closer to the clean data. Second, we find that the purified images are heavily influenced by the internal randomness of diffusion models. When the randomness of the diffusion model is fixed, diffusion models substantially compress the image space. Importantly, we discover a lawful relationship between the adversarial robustness gain and the model's ability to compress the image space, quantified by the expected compression rate (CR). Further theoretical analyses show that (i) convergent score fields encoded in diffusion models explain these compression effects, and (ii) under a low-dimensional data manifold hypothesis, the expected CR captures the compression along off-manifold directions. Our findings uncover the precise mechanisms underlying diffusion-based purification and offer guidance for developing more effective and principled adversarial purification systems.

## 1 INTRODUCTION

Neural networks are vulnerable to small adversarial perturbations (Szegedy et al., 2013; Goodfellow et al., 2014). The lack of robustness presents a fundamental problem of artificial learning systems. Adversarial training (Madry et al., 2017) has been proposed as an effective method to overcome this problem under certain scenarios (Shafahi et al., 2019; Pang et al., 2020; Wang et al., 2021). However, research has found that training with a specific attack usually sacrifices the robustness against other types of perturbations (Schott et al., 2018; Ford et al., 2019; Yin et al., 2019), indicating that adversarial training overfits the attack rather than achieving an overall robustness improvement.

Adversarial purification represents an alternative promising path toward adversarial robustness. This approach typically uses generative models to purify the image before passing it to a classifier (Song et al., 2018; Samangouei et al., 2018; Shi et al., 2021; Yoon et al., 2021). The basic idea is to leverage the image priors learned in generative models to project adversarial perturbations back toward the image manifold. Intuitively, the performance of such purification procedures should depend on how well the generative models encode the probability distribution over natural images. Recently, adversarial purification based on diffusion models (Ho et al., 2020; Song et al., 2020b) (DiffPure) was reported to show impressive improvements against various empirical attacks (Nie et al., 2022). The idea of using diffusion models as denoisers was further combined with the deonise smoothing framework (Cohen et al., 2019; Salman et al., 2020) to improve certificated robustness (Carlini et al., 2022; Xiao et al., 2023). However, more recent work (Lee & Kim, 2023; Li et al., 2025) show that there was an overestimate of the robustness improvement from the DiffPure method. Crucially, despite some promising empirical results, the precise mechanisms underlying the improvement in empirical robustness from diffusion models were still poorly understood.

To close this important gap, we systematically investigated how diffusion models improve adversarial robustness. In this paper, we report a set of surprising phenomena of diffusion models, and identify the key mechanisms for robustness improvements under diffusion-model-based adversarial purification. Our main contributions are summarized below:

- **Revealing intriguing behaviors of diffusion-based purification.** Surprisingly, we find that diffusion models increase—rather than decrease—$\ell_p$ distances to clean samples (Sec.3). Randomness dominates the behavior of diffusion models, where a compression effect emerging once randomness is controlled (Sec.4).

- **Demystifying diffusion-based adversarial purification through a compression framework.** We develop a compression framework to explain the net robustness gain after controlling stochasticity of diffusion models, where the expected compression rate ($\overline{\text{CR}}$) links compression and robustness through a sigmoidal relation (Sec. 5).

- **Deriving new theoretical insights of why diffusion models lead to a compression of image space.** We analyze compression in both Gaussian and diffusion models, showing that it originates from the convergent score field; $\overline{\text{CR}}$ captures off-manifold compression while on-manifold perturbations remain largely preserved (Sec. 6).

## 2 RELATED WORK AND PRELIMINARIES

**Generative models for adversarial purification.** Unlike adversarial training which directly augments the classifier training with adversarial attacks, adversarial purification intends to first "purify" the perturbed image before classification. Generative models are usually utilized as the purification system, such as denoising autoencoder (Gu & Rigazio, 2014), denoising U-Net (Liao et al., 2018), PixelCNN (Song et al., 2018) and GAN (Samangouei et al., 2018). Diffusion models (Ho et al., 2020; Song et al., 2020b) set the SOTA performances on image generation, and represent a natural choice for adversarial purification. Nie et al. (2022) proposed the DiffPure framework, which utilized both the forward and reverse process and achieved promising empirical robustness comparable with adversarial training on multiple benchmarks. Similar improvements were reported with guided diffusion models (Wang et al., 2022). These studies led to substantial interest in applying diffusion models for adversarial purification in various domains, including auditory data (Wu et al., 2022) and 3D point clouds (Sun et al., 2023). Recently, other techniques, such as adversarial guidance (Lin et al., 2024) and bridge models (ADBM) (Li et al., 2025), were introduced to further enhance robustness. Another line of research applies diffusion models to improve certified robustness Cohen et al. (2019). Carlini et al. (2022) found that plugging diffusion models as a denoiser into the denoised smoothing framework (Salman et al., 2020) can lead to non-trivial certified robustness. Xiao et al. (2023) further developed this method and studied the improvement in certified robustness.

**Empirical evaluation of the robustness in diffusion models.** Such randomness may raise concerns about gradient masking in robustness evaluation (Papernot et al., 2017), which provides a false sense of robustness against gradient-based attacks (Tramèr et al., 2018). Athalye et al. (2018) further identified that randomness could cause gradient masking as "stochastic gradients", and proposed the expectation-over-transformation (EOT) which became the standard evaluation for stochastic gradients (Carlini et al., 2019). Additionally, under the assumption that purification systems bring adversarial examples close to clean data, Backward Pass Differentiable Approximation (BPDA) (Athalye et al., 2018) was introduced as a method for evaluating purification-based defenses. However, the proper treatment of randomness in robustness evaluation remains a subject of debate (Gao et al., 2022; Yoon et al., 2021).

In diffusion models, internal randomness and the substantial computational overhead of full-gradient computation make robustness evaluation particularly difficult. The original DiffPure paper applied AutoAttack (Croce & Hein, 2020) with augmented SDE-based gradient estimation and reported a robust accuracy of 70.64% on CIFAR-10. However, through a comprehensive experimental evaluation, (Lee & Kim, 2023) found that the robustness improvements from diffusion models were over-estimated. They recommended using the PGD-EOT with full gradients directly, and estimated the robustness around 46.84%. Li et al. (2025) also challenged the original evaluation and reported a comparable robustness estimate of 45.83%. Recently, Liu et al. (2025) proposed to evaluate the robustness under a deterministic white box setting (DW-box), discovering that the robustness of diffusion models further decreases to 16.8% after controlling the stochasticity.

**Notations and preliminaries.** Denote $\boldsymbol{x}_0$ as the clean image, and $\boldsymbol{x}$ as its perturbed version, so that

$$\boldsymbol{x} = \boldsymbol{x}_0 + \epsilon\boldsymbol{\eta}, \tag{1}$$

where $\boldsymbol{\eta}$ is the normalized adversarial perturbation, and $\epsilon$ controls the magnitude of the attack. Further denote $f$ as the purification system and $g$ as the readout classifier. Adversarial purification typically

consists of two steps: (i) purifying the perturbed image using $f$; (ii) classifying the output using $g$:

$$\hat{\boldsymbol{x}} = f(\boldsymbol{x}), \quad \boldsymbol{y} = g(\hat{\boldsymbol{x}}). \tag{2}$$

Importantly, the purification system may be stochastic. In particular, this is true for diffusion-model-based purification. Diffusion models consist of forward diffusion and reverse denoising processes. The forward process of Denoising Diffusion Probabilistic Models (DDPM) (Ho et al., 2020) is

$$\boldsymbol{x}_t = f_t^{\text{FWD}}(\boldsymbol{x}_{t-1}, \boldsymbol{\epsilon}_t) = \sqrt{\alpha_t}\boldsymbol{x}_{t-1} + \sqrt{1-\alpha_t}\boldsymbol{\epsilon}_t, \ \ \boldsymbol{\epsilon}_t \sim \mathcal{N}(\boldsymbol{0}, \boldsymbol{I}), \tag{3}$$

in which the $\boldsymbol{\epsilon}$ will introduce randomness. Further, the reverse process

$$\boldsymbol{x}_{t-1} = f_t^{\text{REV}}(\boldsymbol{x}_t, \boldsymbol{z}_t) = \frac{1}{\sqrt{\alpha_t}}\left(\boldsymbol{x}_t - \frac{1-\alpha_t}{\sqrt{1-\bar{\alpha}_t}}\boldsymbol{\epsilon}_\theta(\boldsymbol{x}_t, t)\right) + \sigma_t\boldsymbol{z}_t, \ \ \boldsymbol{z}_t \sim \mathcal{N}(\boldsymbol{0}, \boldsymbol{I}) \tag{4}$$

also introduces randomness through $\boldsymbol{z}$. Notably, deterministic reverse process has been proposed, *e.g.,* in Denoising Diffusion Implicit Models (DDIM) (Song et al., 2020a), the reverse process is fully deterministic and thus does not involve randomness.

Denote $\boldsymbol{\xi}$ as a *randomness configuration*, which consists the series of random noises governing the stochastic process, i.e., for DDPM, $\boldsymbol{\xi}^{\text{DDPM}} = \{\boldsymbol{\epsilon}_1 \ldots \boldsymbol{\epsilon}_{t^*}, \boldsymbol{z}_{t^*} \ldots \boldsymbol{z}_1\}$. Denote $f_{\boldsymbol{\xi}} = f(\boldsymbol{x}|\boldsymbol{\xi})$ as the *deterministic* purification conditioned on a particular randomness configuration $\boldsymbol{\xi}$.

## 3 BEHAVIORS OF DIFFUSION MODELS CHALLENGE INTUITIVE HYPOTHESES

While the exact mechanisms underlying robustness improvements from diffusion models remain unclear, prior work has proposed intuitive explanations for how diffusion models may improve robustness. Below we empirically test two such hypotheses.

**(A) "Clean image attraction" hypothesis.** This hypothesis proposes that diffusion models improve robustness by purifying an adversarially perturbed image to be "closer" to the clean image, i.e.,

$$\|f(\boldsymbol{x_0} + \epsilon\boldsymbol{\eta}) - x_0\| \le \epsilon\|\boldsymbol{\eta}\|. \tag{5}$$

The original DiffPure paper (Nie et al., 2022) indicated this view, noting that *"we observe the purified images match the clean images"* and *"recovering clean images from the adversarial examples"*. Intuitively, if a purification system consistently reduces $\ell_p$ distances to the clean image, it effectively transforms an adversarial perturbation into one of smaller magnitude, thereby enhancing robustness.

**(B) "Distribution alignment" hypothesis.** A more sophisticated hypothesis is that, after purification, the distribution of perturbed images become more closely aligned with the clean image distribution,

$$D_{\text{KL}}[p(f(\boldsymbol{x})), p(\boldsymbol{x}_0)] \le D_{\text{KL}}[p(\boldsymbol{x}), p(\boldsymbol{x}_0)], \tag{6}$$

thereby improving robustness. This perspective is also reflected in Theorem 3.1 of Nie et al. (2022). It implies a monotonic relationship between distributional distance and robustness—distributions with smaller KL divergence from the clean distribution are expected to yield higher robustness.

### 3.1 DIFFUSION MODELS PURIFY IMAGES FURTHER AWAY FROM CLEAN IMAGES

**Diffusion models increase $\ell_p$ distances to clean images after purification.** To test whether diffusion models reduce the distance between adversarial and clean images, we conducted a series of experiments. From a clean image, we generated an adversarial example, then applied adversarial purification via diffusion models, and finally measured the distance between the purified image and the original clean image. Surprisingly, we found that the $\ell_2$ distance to the clean sample increased after purification (Fig. 1c). This phenomenon was consistent across a wide range of settings, including different attack types (BPDA, BPDA-EOT, PGD, PGD-EOT), distance metrics ($\ell_2, \ell_\infty$), sampling methods (DDPM, Reverse-only, DDIM) and datasets (CIFAR-10 and ImageNet). It is not specific to adversarial attacks and also holds for perturbations with uniform noise (Appendix E.1, E.2).

**Diffusion-based purification leads to perceptually dissimilar outputs.** While $\ell_p$ distances are highly relevant distance metrics (especially since adversarial attacks are typically defined within bounded $\ell_p$ balls), these metrics may not capture perceptual similarity. For example, translating an image by a single pixel can yield a large $\ell_2$ difference while remaining perceptually identical. Thus, we next ask if diffusion models produce outputs that are perceptually closer to the clean image, even if $\ell_p$ distances increase. To investigate this, we evaluated the structural similarity index measure (SSIM) (Wang et al., 2004), a popular metric used in computer vision for quantifying perceptual similarity of images. As shown in Tables1, we observed a substantial decrease in SSIM between purified and clean images. This indicates that the purified images are not only farther away in $\ell_p$ distances, but also perceptually more dissimilar than the initial adversarial perturbations.

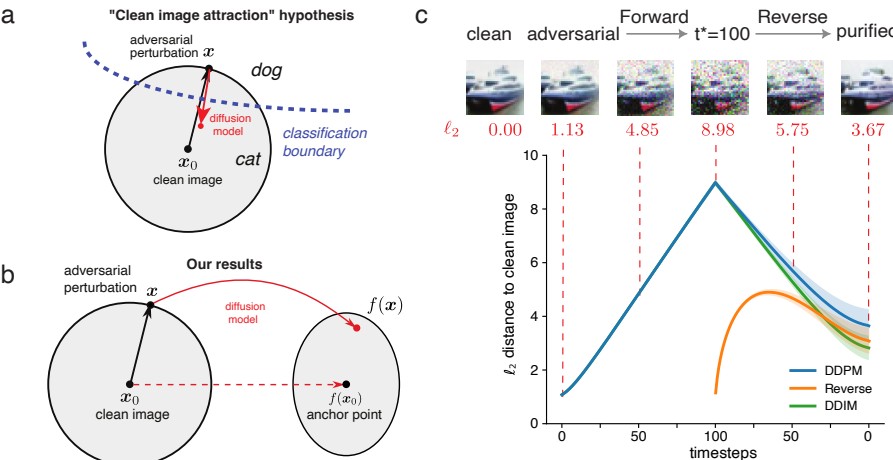

Figure 1: **Diffusion models purify states away from the clean images.** (a) Schematic showing a common hypothesis that diffusion models improve robustness by "denoising" inputs toward the clean image. (b) Summary of our findings, which challenge the denoising hypothesis. (c) Measured $\ell_2$ distances to clean images on CIFAR-10 during purification. We track the distances between intermediate purified states and clean images, using PGD attacks ($\ell_\infty = 8/255$) as initialization. Across all methods, the purified outputs are consistently farther away from the clean image. Additional examples of adversarial and purified images can be found in Appendix H.

Table 1: Distances pre/post-diffusion models on CIFAR-10 (PGD, $\ell_\infty = 8/255$).

| Sampling | $\ell_2$ ($\downarrow$) | $\ell_\infty$ ($\downarrow$) | SSIM ($\uparrow$) |
|---|---|---|---|
| DDPM | $1.077 \rightarrow 3.641$ | $0.031 \rightarrow 0.316$ | $0.965 \rightarrow 0.796$ |
| Reverse | $1.149 \rightarrow 3.078$ | $0.031 \rightarrow 0.270$ | $0.965 \rightarrow 0.837$ |
| DDIM | $1.080 \rightarrow 2.810$ | $0.031 \rightarrow 0.242$ | $0.964 \rightarrow 0.869$ |

### 3.2 DISTRIBUTIONAL DISTANCE FAILS TO EXPLAIN ROBUSTNESS IMPROVEMENTS

We next turn to distribution-level comparisons using the Fréchet Inception Distance (FID) (Heusel et al., 2017). The FID score has been widely used to quantify the performance of generative models such as diffusion models. Here, we measure FID between the adversarial dataset and the clean dataset, both before and after purification, to quantify whether diffusion models bring the distribution of adversarial images closer to that of the clean samples. Interestingly, we observe that purification with diffusion models leads to a reduction of the FID score between adversarial and clean distributions. This is consistent with the idea that diffusion models may bring the distribution of adversarial images closer to the clean data distribution (Li et al., 2025; Nie et al., 2022).

**Non-monotonic relation between distributional distances and adversarial robustness.** We argue that FID is not a reliable indicator of robustness. To illustrate this, we measured the FID distance between the purified adversarial samples and the clean samples across multiple purification methods and timesteps. As shown in Fig. S5a, there is no consistently monotonic relationship between distributional distances and adversarial robustness, suggesting that the distributional alignment hypothesis alone cannot fully explain the observed robustness gains. Distance in the semantic space also fails to explain robustness improvements (Appendix E.3).

Taken together, our empirical results challenge the hypothesis that diffusion models improve robustness by pushing adversarial images closer to their original clean images, either in the $\ell_p$ sense or in the distributional sense.

## 4 DIFFUSION MODELS COMPRESS IMAGE SPACE WITH FIXED RANDOMNESS

Since diffusion models are inherently stochastic, it is instrumental to investigate how much this intrinsic noise affects the output of diffusion models. Specifically, the variability of the output of diffusion models in adversarial purification arises from two distinct sources: (i) the variability in the

input images from perturbations $\boldsymbol{\eta}$ (causing variability around the anchor point), and (ii) the internal variability inherent to the purification system (causing variability of the anchor point).

Fig. 2 illustrates the basic ideas. First, when fixing the input image while allowing the noise in diffusion models to vary, the output variability is solely due to the internal variability of the purification system (Fig. 2a). Surprisingly, we find that this variability is large, as indicated by the relatively low correlations of different purification directions induced by noise (Fig. 2d; mean = 0.22±0.003). Second, when fixing noise in diffusion models and allowing input images to vary (different samples from the image neighborhood), the output variability is solely induced by the variability in the input images (Fig. 2b; mean = 0.93±0.0002). Interestingly, this variability is rather small, as demonstrated in the high correlations between the purified directions (Fig. 2e). By treating input variability as the signal and internal randomness as the noise, we define a signal-to-noise ratio (SNR) of the diffusion purification (Appendix A.4):

$$\text{SNR} = \frac{\mathbb{E}_{\boldsymbol{\xi}}\left[\text{Var}_{\boldsymbol{x}} f(\boldsymbol{x}|\boldsymbol{\xi})\right]}{\text{Var}_{\boldsymbol{\xi}}\left[\mathbb{E}_{\boldsymbol{x}} f(\boldsymbol{x}|\boldsymbol{\xi})\right]}. \tag{7}$$

Numerically evaluating the SNR of diffusion models based on image neighborhood consistent with adversarial attack, we find the SNR is extremely low, i.e., $5.93\pm1.07\times10^{-3}$, indicating that the effect of internal stochasticity is approximately 170 times larger than that induced by input variability.

**Diffusion models compress image space when randomness is fixed.** To investigate these effects further, we measured the $\ell_2$ distances between purified samples and their respective centroids. When starting from the same clean image, internal randomness leads to an expansion effect, where the $\ell_2$ radius of the perturbation space increases from 1.004±0.001 to 3.282±0.453 after purification. Furthermore, the histogram of $\ell_2$ distances shown in Fig. 2f illustrates that varying both the image and noise yields nearly the same distance distribution as varying only the noise. Importantly, we observed the *opposite effect* when randomness in diffusion models is fixed. In this case, the purified outputs become tightly clustered, indicating a strong *compression effect*: the $\ell_2$ radius shrinks to 0.241±0.032. It is useful to note that compression in this paper refers to a reduction in volume, which should not be confused with the compression of data with fewer bits.

**Evaluating the robustness of diffusion models without stochasticity.** The results above indicate that the internal randomness of diffusion models largely determines the final purified output. Con-

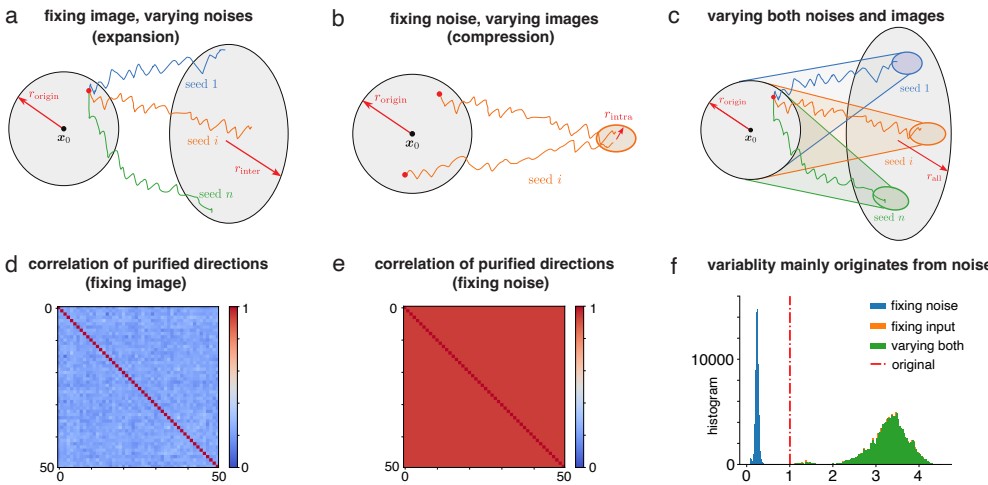

Figure 2: **Behaviors under stochasticity in diffusion models.** (a–c) Schematics illustrating how diffusion models transform input perturbations under different sources of variability. (a) When the image is fixed and internal noise varies, purification exhibits an expansion of the input space. (b) When the noise is fixed and the image varies, the input space are compressed toward a shared direction. (c) When both image and noise vary, internal randomness dominates, producing an overall expansion. (d) Purification directions under different noise samples for the same image are weakly aligned (mean correlation: 0.22±0.003). (e) Under fixed noise, purification directions across perturbed images are highly consistent (mean correlation: 0.93±0.0002). (f) Distribution of $\ell_2$ distances to the centroid. Fixed-noise purification compresses the input ball (radius: $1.004 \rightarrow 0.241$); varying noise leads to expansion (radius: $1.004 \rightarrow 3.282$), confirming that internal randomness drives the dominant effect.

sequently, randomness can substantially influence empirical evaluations of adversarial robustness. We therefore propose that robustness should be assessed under a fixed randomness configuration, which reflects the intrinsic robustness of the model without the confounding effects of stochastic gradients. A concurrent work argued for a similar evaluation protocol (DW-Box (Liu et al., 2025)) under slightly different model settings. Beyond their analysis, our results further suggest that in the presence of randomness, EOT should instead be understood as a *transfer attack* (Appendix C).

In our study, we control the internal randomness of diffusion models by fixing random seeds in both the forward and reverse processes (see Appendix D for implementation details). Once the effects of stochasticity are removed, the robustness gains drop substantially: on CIFAR-10, robust accuracy under PGD attacks falls to 23.7%, and on ImageNet, BPDA with fixed randomness yields 29.5%.[1] These values are markedly lower than previously reported robustness estimates (Tables S2 and S3), yet the robustness improvements from diffusion purification remain non-trivial (*i.e.*, substantially larger than 0). Notably, controlling randomness does not affect clean accuracy. In the remainder of the paper, we focus on understanding robustness improvements in diffusion models without randomness.

## 5    UNDERSTANDING ADVERSARIAL PURIFICATION THROUGH COMPRESSION

Next, we will show how the observed compression effect leads to non-trivial robustness improvements even under fixed randomness. We first introduce our compression theory of adversarial purification, and then demonstrate that compression and adversarial robustness are closely connected through the notion of expected compression rates ($\overline{\text{CR}}$).

### 5.1    A COMPRESSION FRAMEWORK OF ADVERSARIAL PURIFICATION

Following the notations in Sec. 2, for randomness configuration $\boldsymbol{\xi}$, consider the Taylor expansion of $f_{\boldsymbol{\xi}}(\boldsymbol{x})$ around $\boldsymbol{x}_0$:

$$f_{\boldsymbol{\xi}}(\boldsymbol{x}) = f_{\boldsymbol{\xi}}(\boldsymbol{x}_0 + \epsilon\boldsymbol{\eta}) = \underbrace{f_{\boldsymbol{\xi}}(\boldsymbol{x}_0)}_{\text{anchor point}} + \epsilon\underbrace{J_{f_{\boldsymbol{\xi}}}(\boldsymbol{x}_0)}_{\text{compression}}\boldsymbol{\eta} + o(\epsilon), \tag{8}$$

where $J_{f_{\boldsymbol{\xi}}}(\boldsymbol{x}_0)$ is the Jacobian matrix quantifies the local linear transformation induced by $f_{\boldsymbol{\xi}}$.

**Purified clean image $f_{\boldsymbol{\xi}}(\boldsymbol{x}_0)$ as the anchor point.** We observe that the purified clean image $f_{\boldsymbol{\xi}}(\boldsymbol{x}_0)$ naturally defines the center of compression, which we refer to as the *anchor point*. As discussed in Sec. 3, in contrast to the clean image attraction hypothesis, diffusion models do not satisfy $f(\boldsymbol{x}_0) = \boldsymbol{x}_0$. This discrepancy induces a slight drop in clean accuracy, depending on the purification step $t^*$ (TablesS2, S3). Furthermore, as analyzed in Sec.4, each anchor point is uniquely determined by the specific randomness configuration $\boldsymbol{\xi}$.

**Compression rates under infinitesimal isotropic noise.** To quantify the magnitude of the compression effect, we define a scalar measure, the compression rate (CR) of a perturbation vector:

$$\text{CR}[f_{\boldsymbol{\xi}}, \boldsymbol{x}_0, \boldsymbol{\eta}] = \frac{\|f_{\boldsymbol{\xi}}(\boldsymbol{x}_0 + \boldsymbol{\eta}) - f_{\boldsymbol{\xi}}(\boldsymbol{x}_0)\|}{\|\boldsymbol{\eta}\|}, \tag{9}$$

which is evaluated for a specific perturbation $\boldsymbol{\eta}$. One can further define the expected compression rate ($\overline{\text{CR}}$) for perturbations sampled from infinitesimal isotropic noise (i.e., $\mathbb{E}[\boldsymbol{\eta}] = \boldsymbol{0}$, $\mathbb{E}[\boldsymbol{\eta}\boldsymbol{\eta}^T] = I_d$):

$$\overline{\text{CR}}[f_{\boldsymbol{\xi}}, \boldsymbol{x}_0] = \mathbb{E}_{\boldsymbol{\eta}}\text{CR}[f_{\boldsymbol{\xi}}(\boldsymbol{x}_0), \epsilon\boldsymbol{\eta}], \quad \epsilon \to 0, \ \boldsymbol{\eta} \sim \text{Iso}(d). \tag{10}$$

This quantity captures the average local compression effect of $f_{\boldsymbol{\xi}}$ around $\boldsymbol{x}_0$. In practice, we sample uniform noise $\boldsymbol{\eta}$ with the same scale as the adversarial perturbation budget $\epsilon$, thereby matching the $\ell_p$ attack setting. As we will show below, $\overline{\text{CR}}$ is closely connected to adversarial robustness. Importantly, because it is defined as an expectation rather than tied to a particular $\boldsymbol{\eta}$, $\overline{\text{CR}}$ can be estimated easily using sampling, without expensive gradient computations.

### 5.2    RELATION BETWEEN COMPRESSION RATES AND ADVERSARIAL ROBUSTNESS

Adversarial examples arise when a small neighborhood around clean image intersects the classifier's decision boundary. We reason that the compression of image space effectively reduces the size of image neighborhood and thus reduces the odds of the transformed neighborhood intersecting with a decision boundary. Thus, we predict that there should be a direct relationship between the compression rate and adversarial robustness.

---

[1]We were unable to compute full PGD gradients for diffusion models on ImageNet.

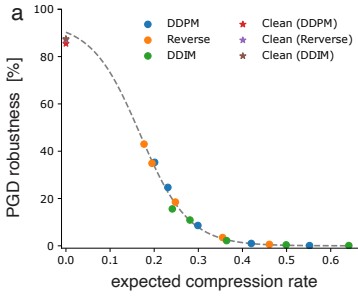 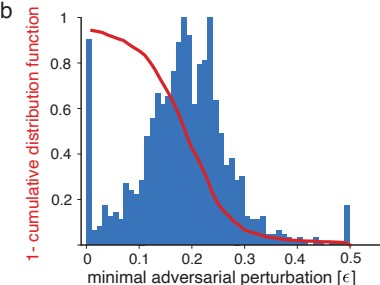

Figure 3: **Lawful relation between expected compression rates and adversarial robustness.** (a) The compression rate and robustness without stochasticity of diffusion models follow a consistent relation well captured by a sigmoid function. Note that the curve generalizes across different sampling methods and extrapolates smoothly to clean accuracies at the y-intercept. (b) The sigmoidal relation arises from the survival function $(1 - \text{CDF})$ of the distribution of minimum adversarial perturbations of the classifier, an intrinsic property of the classifier that is independent of purification systems. Note that the distribution at the clean samples is qualitatively similar to that at the anchor points (Fig. S1).

**Lawful relation between compression rate and adversarial robustness.** We test this predicted relation using CIFAR-10. We take advantage of the fact that different implementations of diffusion models (DDPM, Reverse-only, DDIM) lead to different adversarial robustness (Table 2). Examining a collection of diffusion models with various hyperparameters, we plot their robustness performances against their compression rates. Strikingly, we find that the two quantities exhibit a lawful relationship (Fig. 3a). The compression-robustness curve increases sharply for a compression rate around 0.2. We find that the relation between compression and robustness can be well fitted by a sigmodal function. Interestingly, extrapolating the fitted curve to zero compression rate leads to a relatively accurate prediction of the clean accuracy.

Table 2: The expected compression rates and adversarial robustness on CIFAR-10.

| Model | Metric | $t = 10$ | $t = 20$ | $t = 50$ | $t = 100$ | $t = 150$ |
|---|---|---|---|---|---|---|
| DDPM (Ho et al., 2020) | $\overline{\text{CR}}$ | 0.552 | 0.420 | 0.299 | 0.231 | 0.201 |
| | PGD-Fix | 0.1% | 1.0% | 8.6% | 23.7% | 35.3% |
| Reverse-only DDPM (Xiao et al., 2023) | $\overline{\text{CR}}$ | 0.461 | 0.355 | 0.248 | 0.195 | 0.177 |
| | PGD-Fix | 0.6% | 3.5% | 18.5% | 34.9% | 43.0% |
| DDIM (Song et al., 2020a) | $\overline{\text{CR}}$ | 0.641 | 0.499 | 0.364 | 0.281 | 0.241 |
| | PGD-Fix | 0.1% | 0.4% | 2.2% | 10.9% | 15.6% |

**Sigmoidal relation arises from the distribution of minimum perturbation of the classifier.** The observed sigmoidal relationship between compression rate and robustness points to a more fundamental explanation. Building on our compression-based theory, we hypothesize that this pattern reflects the intrinsic link between attack budget and robustness. To investigate, we estimate the distribution of minimum perturbation magnitudes required to fool the CIFAR-10 classifier by performing a binary search over perturbation scales $\epsilon$ for each test image. As shown in Fig. 3b, the resulting distribution is well described by a zero-inflated Gaussian: the Gaussian component captures the bulk of perturbation thresholds, while the zero-inflated component reflects clean input errors, for which any perturbation suffices. This characterization explains the sigmoidal robustness curve—the *survival function* $(1 - \text{CDF})$ of such a distribution naturally yields a sigmoid, with the plateau determined by the classifier's clean accuracy. These findings suggest that the sigmoidal relation arises from the intrinsic distribution of minimum perturbations imposed by the classifier's decision boundary, rather than from properties of the purification system itself. Consequently, different purification methods (e.g., DDPM, reverse-only, DDIM) fall on the same curve because they compress perturbations to varying degrees but are governed by the same classifier-induced distribution. Moreover, this perspective provides a principled way to predict adversarial robustness from compression rates alone, without requiring gradient-based evaluation.

## 5.3 ESTIMATING THE ADVERSARIAL ROBUSTNESS WITH COMPRESSION RATES

We next investigate whether one can go one step further by using the compression rate (CR) and the minimal adversarial perturbation of the base classifier to predict the robustness of a purification system. For a given purification system, we can compute (i) the expected CR, and (ii) the survival function for the base classifier at the anchor points (akin to red curve in Fig. 3b, see Appendix B). Using (ii), we identify the survival probability corresponds to the CR from (i), and use this survival probability as the predicted adversarial robustness.

We investigate how well such predictions based on CR match empirical robustness for CIFAR-10. As shown in Table 3, theoretical predictions are generally consistent with the empirically measured robustness. They are more accurate for DDIM than the other two samplers. The observed deviation may be due to multiple reasons. First, the survival function (see red curve in Fig. 3b) changes rapidly around CR = 0.2. Second, the CR of the image neighborhood is not uniform for all perturbations.

We conduct further analysis to generate predictions on the ImageNet dataset using DDPM and two additional continuous-time sampling methods (VPSDE from Song et al. (2020b), and DPM from Lu et al. (2022)). Across all diffusion models, substantial compression of the image space were observed (Table 4). Based on these measured CR, our theory predicts that DDPM should exhibit slightly higher robustness than VPSDE, and both of them should substantially outperform DPM-20 in robustness. These provide testable predictions for future work.

Table 3: Compression rates and robustness predictions on CIFAR-10 ($t = 100$).

| Method | Clean Acc. | Expected CR | Pred. (Clean) | Pred. (Anchor) | PGD Robust. |
|---|---|---|---|---|---|
| DDPM | 85.5% | 0.231±0.046 | 27.8% | 26.0% | 23.7% |
| Reverse | 87.2% | 0.195±0.046 | 45.3% | 40.9% | 34.9% |
| DDIM | 87.5% | 0.281±0.048 | 10.2% | 10.6% | 10.9% |

Table 4: Compression rates and robustness predictions on ImageNet ($t = 100$).

| Method | Clean Acc. | Expected CR | Pred. (Clean) | Pred. (Anchor) |
|---|---|---|---|---|
| DDPM (Ho et al., 2020) | 73.6% | 0.147±0.070 | 46.1% | 42.2% |
| VPSDE (Song et al., 2020b) | 75.4% | 0.165±0.059 | 40.1% | 38.0% |
| DPM-20 (Lu et al., 2022) | 59.8% | 0.337±0.015 | 1.0% | 0.5% |

**Reliability analysis of the expected compression rates.** We next analyze the reliability of the expected compression rates. Given the simplicity of the definition, the only hyperparameter involved is the magnitude of the random perturbation $\epsilon$. When $\epsilon$ is sufficiently small to make the first-order Taylor expansion valid, the exact magnitude of $\epsilon$ should not matter. Empirically, we used the standard magnitude of the adversarial attack for each dataset. To further examine the relationship between perturbation magnitude and CR, we measured the expected CR ranging from 1/4 to 4 times of the original adversarial magnitude on CIFAR-10. As shown in Table 5, for small $\epsilon$ (up to 16/255), CR remains stable. Only when $\epsilon$ becomes noticeably large (e.g., 32/255), the CR begins to drift. This is expected as the Taylor approximation would break down for large perturbations. We also verified that the expected CR is stable across different random seeds (Table 6). Together, these results suggest that the expected CR is a simple yet remarkably stable quantity that reflects the intrinsic compression capability of diffusion models.

Table 5: Compression rates with different perturbation scales (DDPM, CIFAR-10).

| Epsilon | 2 / 255 | 4 / 255 | 8 / 255 | 16 / 255 | 32 / 255 |
|---|---|---|---|---|---|
| Expected CR | 0.230±0.047 | 0.230±0.046 | 0.231±0.046 | 0.239±0.049 | 0.268±0.057 |

Table 6: Compression rates across random seeds (DDPM, CIFAR-10).

| Seed | 0 | 123 | 295 |
|---|---|---|---|
| Expected CR | 0.2306±0.0457 | 0.2309±0.0481 | 0.2312±0.0481 |

# 6 THEORETICAL ANALYSIS OF THE COMPRESSION PROCESS

## 6.1 THE COMPRESSION PROCESS IN A GAUSSIAN SCORE FIELD

We next perform theoretical analyses to understand the diffusion-model-induced compression effect reported above. To build intuition, we first examine the Gaussian score field. Assume the data $\boldsymbol{x} \in \mathbb{R}^d$ follow the multivariate Gaussian distribution, $\boldsymbol{x} \sim \mathcal{N}(\boldsymbol{\mu}, \Sigma)$, where $\Sigma$ is the covariance matrix (symmetric and positive definite). The score function

$$s(\boldsymbol{x}) = \nabla_{\boldsymbol{x}} \log p(\boldsymbol{x}) = -\Sigma^{-1}(\boldsymbol{x} - \boldsymbol{\mu}). \tag{11}$$

Assume a dynamical system follows the score field, and discretize with the forward Euler method,

$$\frac{d\boldsymbol{x}}{dt} = s(\boldsymbol{x}) \implies \boldsymbol{x}_{t+1} = f_t(\boldsymbol{x}_t) = \boldsymbol{x}_t - h_t \cdot \Sigma^{-1}(\boldsymbol{x}_t - \boldsymbol{\mu}), \tag{12}$$

where $h_t$ is the step size at timestep $t$. Suppose the covariance matrix $\Sigma$ has eigenvector $\boldsymbol{v}_i$ with eigenvalue $\sigma_i^2$, the CR along the eigenvector, and the expected CR are given by (Appendix A.2)

$$\text{CR}[f_t, \boldsymbol{\eta} /\!/ \boldsymbol{v}_i] = 1 - \frac{h_t}{\sigma_i^2}, \quad \overline{\text{CR}}[f_t] \approx 1 - \frac{h_t}{d} \sum_i \frac{1}{\sigma_i^2}. \tag{13}$$

The results show a strong compression effect (small CR) along directions of low variance (off-manifold) and a weaker compression effect (large CR) along directions of high variance (on-manifold; see Fig. 4a).

Natural images are often modeled under the low-dimensional manifold hypothesis, meaning that images lie in a low-dimensional subspace embedded in a high-dimensional space (Simoncelli & Olshausen, 2001; Maaten & Hinton, 2008). For instance, Pope et al. (2021) estimated the intrinsic dimension of CIFAR-100 to be around 30 and ImageNet to be around 45, both much lower than their corresponding embedding dimensions. As a first step toward understanding how the low-dimensionality may affect compression, we study the case that the covariance matrix has an intrinsic dimension $d_{\text{in}} \ll d$, with large variances $\sigma_l^2$ along these intrinsic dimensions (i.e., on-manifold directions) and small variances $\sigma_s^2$ along the remaining ones (i.e., off-manifold directions). That is, the covariance matrix can be diagonalized as $\Lambda = \text{diag}(\sigma_l^2 I_{d_{\text{in}}}, \sigma_s^2 I_{d-d_{\text{in}}})$, with $d_{\text{in}} \ll d$, $\sigma_l^2 \gg \sigma_s^2$. Under these assumptions, we find that the compression rates

$$\text{CR}_{\text{on-manifold}} = \text{CR}[f_t, \boldsymbol{\eta} /\!/ \boldsymbol{v}_l] = 1 - \frac{h_t}{\sigma_l^2}, \quad \text{CR}_{\text{off-manifold}} = \text{CR}[f_t, \boldsymbol{\eta} /\!/ \boldsymbol{v}_s] \approx \overline{\text{CR}}. \tag{14}$$

The dominant compression effect along off-manifold directions is well captured by the expected CR. We illustrate this by empirically tracking the CR over time for a random initial perturbation as the system evolves. As shown in Fig. 4b, the CR begins at the off-manifold value and eventually converges to the on-manifold value, demonstrating that off-manifold perturbations are compressed while on-manifold components are preserved, consistent with the accuracy of our approximation.

## 6.2 THE COMPRESSION PROCESS OF DIFFUSION MODELS

**Compression induced by the convergent score field of reverse process.** We next extend our analyses to diffusion models. Calculate the CR of forward process at timestep $t$ with randomness $\boldsymbol{\epsilon}_t$

$$\text{CR}[f_{t,\boldsymbol{\epsilon}_t}^{\text{FWD}}, \boldsymbol{x}_t, \boldsymbol{\eta}] = \sqrt{\alpha_t} \approx 1. \tag{15}$$

The forward process is primarily a translational shift, producing uniform compression across all directions (Fig. 4c). The total compression induced by the forward process is bounded by $\bar{\alpha}_t = \prod_{t=1}^{t^*} \sqrt{\alpha_t} \approx 0.90$, which is not sufficient to yield substantial robustness gains according to our theory (the robustness gain is prominent only after reaching a CR of around 0.3, Fig. 3b).

Let $\kappa_t = 1/\sqrt{\alpha_t}$, $\gamma_t = (1 - \alpha_t)/(\sqrt{1 - \bar{\alpha}_t})$, the expected CR of the reverse process at timestep $t$ is given by (Appendix A.3),

$$\overline{\text{CR}}[f_{t,\boldsymbol{z}_t}^{\text{REV}}, \boldsymbol{x}_t] \approx \kappa_t \left(1 - \frac{\gamma_t}{d} \nabla \cdot \boldsymbol{\epsilon}_\theta(\boldsymbol{x}_t, t)\right), \tag{16}$$

where $\nabla \cdot \boldsymbol{\epsilon}_\theta(\boldsymbol{x}_t, t) = \text{tr}[J_{\boldsymbol{\epsilon}_\theta}(\boldsymbol{x}_t, t)]$ is the divergent of the noise predictor. In practice, $\kappa_t$ is close to 1. This implies that a compression effect arises when the divergence $\nabla \cdot \boldsymbol{\epsilon}_\theta > 0$. By Tweedie's

formula (Robbins, 1956; Miyasawa et al., 1961), the score function at $\boldsymbol{x}_t$ can be estimated using the optimal noise predictor (Appendix A.1).

$$s(\boldsymbol{x}_t) \approx -\frac{1}{\sqrt{1 - \bar{\alpha}_t}} \boldsymbol{\epsilon}_\theta(\boldsymbol{x}_t, t). \tag{17}$$

Thus, in the reverse process of diffusion models, the compression effect is induced by the convergent nature of the learned score function, $\nabla \cdot s(\boldsymbol{x}_t) < 0$.

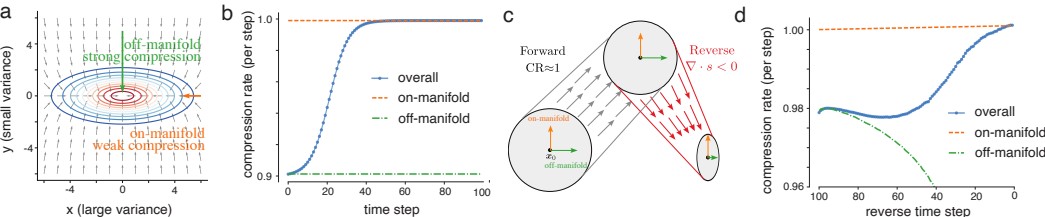

Figure 4: **Theoretical analysis of the compression process.** (a) Score field of a multivariate Gaussian distribution. The Gaussian exhibits constant CR along each direction, with small compression on-manifold and large compression off-manifold. (b) As the system evolves through the Gaussian score field, CR transitions from the off-manifold value to the on-manifold value. (c) Illustration of compression in diffusion models. The reverse process contributes most of the compression effect, primarily acting on off-manifold perturbations. (d) Measured CR at each step of diffusion models. Similar to the Gaussian case, CR transitions from the off-manifold to the on-manifold value. The initial match of the CR (blue) with the off-manifold value (green) supports our approximation in Eq. 16. As the reverse process progresses, the two curves diverge, reflecting the perturbations being purified from off- to on-manifold directions.

**Diffusion model purifies off-manifold while preserves on-manifold perturbations.** Analogous to the Gaussian case, define the eigendirection with small absolute eigenvalue $|\tilde{\lambda}_s|$ of the Jacobian of the score $J_s(\boldsymbol{x})$ as on-manifold, and the large absolute eigenvalue $|\tilde{\lambda}_l|$ direction as off-manifold. Following the low-dimensional data manifold hypothesis, we assume that the eigenvalue matrix has the structure

$$\Lambda_{J_s(\boldsymbol{x})} = \mathrm{diag}\big(\tilde{\lambda}_s I_{d_{\mathrm{in}}}, \tilde{\lambda}_l I_{d - d_{\mathrm{in}}}\big), \quad d_{\mathrm{in}} \ll d, \ |\tilde{\lambda}_s| \ll |\tilde{\lambda}_l|. \tag{18}$$

We can approximate the CR along both on-manifold and off-manifold directions as (Appendix A.3)

$$\mathrm{CR}_{\text{on-manifold}} \approx \kappa_t, \quad \mathrm{CR}_{\text{off-manifold}} \approx \overline{\mathrm{CR}}[f_{t,\boldsymbol{z}_t}^{\mathrm{REV}}, \boldsymbol{x}_t]. \tag{19}$$

We again empirically validate our theory by tracking the CR over time for a random initial perturbation under diffusion purification. As shown in Fig.4d, similar to the Gaussian case, during diffusion purification the CR initially reflects off-manifold value and gradually approaches the on-manifold value, indicating that off-manifold perturbations are selectively compressed while on-manifold components are preserved. The initial match of the CR (blue) with the off-manifold value (green) further supports the accuracy of our first-order approximation in Eq. 16. As the reverse process progresses, the two curves diverge because the perturbations are transformed from off- to on-manifold directions.

## 7 DISCUSSIONS

We have systematically analyzed how diffusion models improve adversarial robustness. We find that diffusion models push perturbed images away from clean samples while simultaneously compressing the image space around the anchor points. Moreover, the compression rate and empirical robustness follow a systematic relationship, providing strong support for the hypothesis that compression of the image space underlies the robustness improvements offered by diffusion models. Our results suggest a promising direction for designing "compression-based purification" systems, which should satisfy two criteria: (i) high clean accuracy at anchor points, and (ii) strong compression rates around anchor points. Systems meeting these criteria are expected to achieve reliable adversarial robustness without relying on stochasticity. In this work, we focused on diffusion models constructed in the image space, as most prior studies on diffusion-based purification have used this type of model (Nie et al., 2022; Xiao et al., 2023; Lee & Kim, 2023; Liu et al., 2025). Recently, latent diffusion models (Rombach et al., 2022) and flow-matching models (Lipman et al., 2023) have also been shown to improve adversarial robustness (Zhang et al., 2025; Collaert et al., 2025). It would be interesting for future work to test whether our findings generalize to these alternative purification models.

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

# A  THEORETICAL RESULTS

## A.1  SCORE FUNCTION OF DDPM VIA TWEEDIE'S FORMULA

*Proof.* The score function of the noisy sample $\boldsymbol{x}_t$ is defined as

$$s(\boldsymbol{x}_t) = \nabla_{\boldsymbol{x}_t} \log p(\boldsymbol{x}_t).$$

By Tweedie's formula (Robbins, 1956; Miyasawa et al., 1961), for a Gaussian observation

$$\boldsymbol{x}_t \sim \mathcal{N}(\boldsymbol{\mu}, \sigma^2 \boldsymbol{I}),$$

the posterior mean $\mathbb{E}[\boldsymbol{\mu} \mid \boldsymbol{x}_t]$ can be expressed in terms of the score function:

$$\mathbb{E}[\boldsymbol{\mu} \mid \boldsymbol{x}_t] = \boldsymbol{x}_t + \sigma^2 \nabla_{\boldsymbol{x}_t} \log p(\boldsymbol{x}_t).$$

In the DDPM forward process (eq. 3), we can rewrite $\boldsymbol{x}_t$ as

$$\boldsymbol{x}_t = \sqrt{\bar{\alpha}_t} \boldsymbol{x}_0 + \sqrt{1 - \bar{\alpha}_t} \, \boldsymbol{\epsilon}, \quad \boldsymbol{\epsilon} \sim \mathcal{N}(\boldsymbol{0}, \boldsymbol{I}),$$

where $\sqrt{\bar{\alpha}_t} \boldsymbol{x}_0$ plays the role of the unknown mean $\boldsymbol{\mu}$ and $\sqrt{1 - \bar{\alpha}_t}$ is the standard deviation.

Applying Tweedie's formula gives

$$\nabla_{\boldsymbol{x}_t} \log p(\boldsymbol{x}_t) = \frac{\mathbb{E}[\sqrt{\bar{\alpha}_t} \, \boldsymbol{x}_0 \mid \boldsymbol{x}_t] - \boldsymbol{x}_t}{1 - \bar{\alpha}_t}.$$

The DDPM reverse process (eq. 4) estimates $\boldsymbol{x}_0$ using the noise predictor $\boldsymbol{\epsilon}_\theta(\boldsymbol{x}_t, t)$:

$$\sqrt{\bar{\alpha}_t} \, \boldsymbol{x}_0 \approx \boldsymbol{x}_t - \sqrt{1 - \bar{\alpha}_t} \, \boldsymbol{\epsilon}_\theta(\boldsymbol{x}_t, t).$$

Plugging this into the score function expression yields

$$\begin{aligned}
s(\boldsymbol{x}_t) &= \nabla_{\boldsymbol{x}_t} \log p(\boldsymbol{x}_t) \\
&\approx \frac{\boldsymbol{x}_t - \sqrt{1 - \bar{\alpha}_t} \, \boldsymbol{\epsilon}_\theta(\boldsymbol{x}_t, t) - \boldsymbol{x}_t}{1 - \bar{\alpha}_t} \\
&= -\frac{1}{\sqrt{1 - \bar{\alpha}_t}} \, \boldsymbol{\epsilon}_\theta(\boldsymbol{x}_t, t).
\end{aligned}$$

Hence, the DDPM noise predictor $\boldsymbol{\epsilon}_\theta(\boldsymbol{x}_t, t)$ provides a direct estimate of the score function up to a scaling factor. $\square$

## A.2  COMPRESSION IN THE GAUSSIAN SCORE FIELD

*Proof.* Let $\boldsymbol{x} \sim \mathcal{N}(\boldsymbol{x}; \boldsymbol{\mu}, \Sigma)$,

$$p(\boldsymbol{x}) = \frac{1}{(2\pi)^{d/2} |\Sigma|^{1/2}} \exp\left( -\tfrac{1}{2} (\boldsymbol{x} - \boldsymbol{\mu})^T \Sigma^{-1} (\boldsymbol{x} - \boldsymbol{\mu}) \right),$$

where $\Sigma$ is a positive-definite covariance matrix.

The score function

$$s(\boldsymbol{x}) = \nabla_{\boldsymbol{x}} \log p(\boldsymbol{x}) = -\frac{1}{2} \nabla_{\boldsymbol{x}} \left[ (\boldsymbol{x} - \boldsymbol{\mu})^T \Sigma^{-1} (\boldsymbol{x} - \boldsymbol{\mu}) \right] = -\Sigma^{-1} (\boldsymbol{x} - \boldsymbol{\mu}).$$

The Jacobian of the score

$$J_s(\boldsymbol{x}) = \frac{\partial}{\partial \boldsymbol{x}} \left[ -\Sigma^{-1} (\boldsymbol{x} - \boldsymbol{\mu}) \right] = -\Sigma^{-1}.$$

which is constant and negative definite.

Since the covariance matrix is symmetric positive definite, diagonalize $\Sigma = V \Lambda V^T$,

$$\Lambda = \mathrm{diag}(\sigma_1^2, \ldots, \sigma_d^2), \ V = (\boldsymbol{v}_1, \ldots, \boldsymbol{v}_d),$$

where $V$ is the orthonormal eigenvector matrix. The inverse of the covariance $\Sigma^{-1}$ can be diagonalized with the same eigenvectors as

$$\Sigma^{-1} = V\Lambda^{-1}V^T, \ \Lambda^{-1} = \mathrm{diag}\left(\frac{1}{\sigma_1^2}, \dots, \frac{1}{\sigma_d^2}\right).$$

Therefore, the compression rate along a certain eigenvector $\boldsymbol{v}_i$ of $\Sigma$ is

$$\mathrm{CR}[f_t, \boldsymbol{\eta}/\!/\boldsymbol{v}_i] = \frac{\|(I_d - h_t\Sigma^{-1})\boldsymbol{\eta}\|}{\|\boldsymbol{\eta}\|} = 1 - \frac{h_t}{\sigma_i^2}.$$

For a perturbation $\boldsymbol{\eta}$ in general, decomposite onto the eigenvectors, $\boldsymbol{\eta} = \sum_i a_i \boldsymbol{v}_i$. Therefore the noise residue (numerator)

$$f_t(\boldsymbol{x}_0 + \boldsymbol{\eta}) - f_t(\boldsymbol{x}_0) = (I_d - h_t\Sigma^{-1})\boldsymbol{\eta} = \sum_i \left(1 - \frac{h_t}{\sigma_i^2}\right) a_i \boldsymbol{v}_i.$$

For isotropic noise, $a_i = 1/d$, $\|\boldsymbol{\eta}\| = \sqrt{1/d}$ is almost constant for large $d$, therefore the expexted CR

$$\overline{\mathrm{CR}} = \sqrt{\sum_i \left(1 - \frac{h_t}{\sigma_i^2}\right)^2 \frac{1}{d}} = \sqrt{1 - 2\frac{h_t}{d}\sum_i \frac{1}{\sigma_i^2} + \frac{h_t^2}{d}\sum_i \frac{1}{\sigma_i^4}}.$$

Since the timestep $h_t \to 0$, $d \to \infty$, $\sqrt{1+u} \approx 1 + u/2$ $(u \to 0)$, consequently

$$\overline{\mathrm{CR}} \approx 1 - \frac{h_t}{d}\sum_i \frac{1}{\sigma_i^2}.$$

Define the direction of large variance $\sigma_L^2$ as the on-manifold direction, and the small variance $\sigma_s^2$ direction as off-manifold. By the low-dimensional manifold assumption, the eigenvalue matrix has the structure

$$\Lambda = \mathrm{diag}\big(\underbrace{\sigma_l^2, \dots, \sigma_l^2}_{d_{\mathrm{in}}}, \underbrace{\sigma_s^2, \dots, \sigma_s^2}_{d-d_{\mathrm{in}}}\big),$$

with $d_{\mathrm{in}} \ll d$, $\sigma_l^2 \gg \sigma_s^2$. Therefore,

$$\mathrm{CR}_{\text{on-manifold}} = \mathrm{CR}[f_t, \boldsymbol{\eta}/\!/\boldsymbol{v}_l] = 1 - \frac{h_t}{\sigma_l^2},$$

$$\mathrm{CR}_{\text{off-manifold}} = \mathrm{CR}[f_t, \boldsymbol{\eta}/\!/\boldsymbol{v}_s] = 1 - \frac{h_t}{\sigma_s^2} \approx 1 - \frac{d - d_{\mathrm{in}}}{d}\frac{h_t}{\sigma_s^2} \approx \overline{\mathrm{CR}}.$$

$\square$

The hyperparameters used to simulate the Gaussian score field in Fig. 4 are listed in Table. S1.

Table S1: Hyperparameters for simulating the Gaussian score field.

| Hyperparameters | Values |
|---|---|
| Dimension $d$ | 3072 |
| Intrinsic dimension $d_{\mathrm{in}}$ | 40 |
| On-manifold std $\sigma_l$ | 1.0 |
| Off-manifold std $\sigma_s$ | 0.01 |
| Stepsize $h_t$ | 0.001 |
| Timesteps $T$ | 100 |

### A.3 COMPRESSION IN DIFFUSION MODELS

*Proof.* DiffPure (Nie et al., 2022) runs the forward and reverse process of the diffusion model to an intermediate $t^*$ step as a purification system,

$$f^{\text{DiffPure}}(\boldsymbol{x}) = f_1^{\text{REV}} \circ \cdots \circ f_{t^*}^{\text{REV}} \circ f_{t^*}^{\text{FWD}} \cdots \circ f_1^{\text{FWD}}(\boldsymbol{x}).$$

Following the definition, calculate the CR of forward process (eq. 3) at timestep $t$ with randomness $\boldsymbol{\epsilon}_t$

$$\text{CR}[f_{t,\boldsymbol{\epsilon}_t}^{\text{FWD}}, \boldsymbol{x}_t, \boldsymbol{\eta}] = \frac{\|\alpha_t \boldsymbol{\eta}\|}{\|\boldsymbol{\eta}\|} = \alpha_t \approx 1.$$

Thus, the forward process is primarily a translational shift, producing slight uniform compression across all directions.

Denote the reverse step of the diffusion model at timestep $t$ (eq. 4) as

$$\boldsymbol{x}_{t-1} = f_t(\boldsymbol{x}_t) = \kappa_t \left( \boldsymbol{x}_t - \gamma_t \boldsymbol{\epsilon}_\theta(\boldsymbol{x}_t, t) \right) + \sigma_t \boldsymbol{z}_t,$$

where $\kappa_t = 1/\sqrt{\alpha_t}$, $\gamma_t = (1 - \alpha_t)/(\sqrt{1 - \bar{\alpha}_t})$. Consider the Jacobian of the transformation $f_t$

$$J_{f_t}(\boldsymbol{x}_t) = \kappa_t(I_d - \gamma_t J_{\boldsymbol{\epsilon}_\theta}(\boldsymbol{x}_t, t)).$$

Suppose the Jacobian of the noise predictor $J_{\boldsymbol{\epsilon}_\theta}(\boldsymbol{x}_t, t)$ has eigenvector $\boldsymbol{v}_i$ with eigenvalue $\lambda_i$, then

$$J_{f_t}(\boldsymbol{x}_t)\boldsymbol{v}_i = \kappa_t(1 - \gamma_t \lambda_i)\boldsymbol{v}_i.$$

At timestep $t$, with fixed randomness $\boldsymbol{z}_t$, the noise residue (numerator) with infinitesimal perturbation $\epsilon \to 0$,

$$f_t(\boldsymbol{x}_t + \epsilon \boldsymbol{\eta}) - f_t(\boldsymbol{x}_t) \approx J_{f_t}(\boldsymbol{x}_t)\epsilon \boldsymbol{\eta}.$$

Therefore the CR along the eigendirection $\boldsymbol{v}_i$,

$$\text{CR}[f_{t,\boldsymbol{z}_t}^{\text{REV}}, \boldsymbol{x}_t, \epsilon \boldsymbol{v}_i] \approx \kappa_t(1 - \gamma_t \lambda_i).$$

For isotropic noise, decomposite $\boldsymbol{\eta}$ onto eigendirections, $\boldsymbol{\eta} = \sum_i a_i \boldsymbol{v}_i$, where $a_i = 1/d$, $\|\boldsymbol{\eta}\| = \sqrt{1/d}$ is almost constant for large $d$. Therefore the expected CR at timestep $t$

$$\overline{\text{CR}}[f_{t,\boldsymbol{z}_t}^{\text{REV}}, \boldsymbol{x}_t] \approx \kappa_t \sqrt{\frac{1}{d} \sum_i (1 - \gamma_t \lambda_i)^2}$$

$$= \kappa_t \sqrt{1 - \frac{2\gamma_t}{d}\left(\sum_i \lambda_i\right) + \frac{\gamma_t^2}{d}\left(\sum_i \lambda_i^2\right)}$$

$$= \kappa_t \sqrt{1 - \frac{2\gamma_t}{d}\nabla \cdot \boldsymbol{\epsilon}_\theta(\boldsymbol{x}_t, t) + \frac{\gamma_t^2}{d}\|J_{\boldsymbol{\epsilon}_\theta}(\boldsymbol{x}_t, t)\|_F^2}.$$

Since $\gamma_t \to 0$, $d \to \infty$, $\overline{\text{CR}}$ can be further approximated as

$$\overline{\text{CR}} \approx \kappa_t \left(1 - \frac{\gamma_t}{d}\nabla \cdot \boldsymbol{\epsilon}_\theta(\boldsymbol{x}_t, t)\right).$$

By Tweedie's formula, $\boldsymbol{v}_i$ is also an eigenvector of the Jacobian of the score function $J_s(\boldsymbol{x}_t)$, with the eigenvalue

$$\tilde{\lambda}_i = -\frac{\lambda_i}{\sqrt{1 - \bar{\alpha}_t}}, \quad \tilde{\boldsymbol{v}}_i = \boldsymbol{v}_i.$$

Define the eigendirection with small absolute eigenvalue $|\tilde{\lambda}_s|$ of the Jacobian of the score $J_s(\boldsymbol{x})$ as the on-manifold direction, and the large absolute eigenvalue $|\tilde{\lambda}_l|$ direction as off-manifold. By the low-dimensional manifold assumption, the eigenvalue matrix has the structure

$$\Lambda_s(\boldsymbol{x}) = \text{diag}\big( \underbrace{\tilde{\lambda}_s, \ldots, \tilde{\lambda}_s}_{d_{\text{in}}}, \underbrace{\tilde{\lambda}_l, \ldots, \tilde{\lambda}_l}_{d - d_{\text{in}}} \big),$$

with $d_{\text{in}} \ll d$, $|\tilde{\lambda}_s| \ll |\tilde{\lambda}_l| \Rightarrow |\lambda_s| \ll |\lambda_l|$. Consequently,

$$\text{CR}_{\text{on-manifold}} = \text{CR}[f_t, \boldsymbol{\eta} /\!/ \boldsymbol{v}_s] \approx \kappa_t(1 - \gamma_t \lambda_s) \approx \kappa_t,$$

$$\text{CR}_{\text{off-manifold}} = \text{CR}[f_t, \boldsymbol{\eta} /\!/ \boldsymbol{v}_l] \approx \kappa_t(1 - \gamma_t \lambda_l) \approx \kappa_t \left(1 - \gamma_t \frac{d - d_{\text{in}}}{d}\lambda_l\right) \approx \overline{\text{CR}}.$$

$\square$

## A.4 VARIANCE DECOMPOSITION IN DIFFUSION MODELS FOR ADVERSARIAL PURIFICATION

Assume inputs $\boldsymbol{x}$ are independent with randomness $\boldsymbol{\xi}$, the total variance of diffusion models can then be decomposed as

$$\text{Var}_{\boldsymbol{x},\boldsymbol{\xi}}[f(\boldsymbol{x},\boldsymbol{\xi})] = \mathbb{E}_{\boldsymbol{\xi}}\left[\text{Var}_{\boldsymbol{x}}f(\boldsymbol{x}|\boldsymbol{\xi})\right] + \text{Var}_{\boldsymbol{\xi}}\left[\mathbb{E}_{\boldsymbol{x}}f(\boldsymbol{x}|\boldsymbol{\xi})\right]. \tag{20}$$

The result directly follows the law of total variance. Here we append the proof for completeness.

*Proof.* Define the following means

$$\mu_{\boldsymbol{\xi}} := \mathbb{E}_{\boldsymbol{x}}f(\boldsymbol{x}|\boldsymbol{\xi}) \quad \text{(mean at fixed randomness } \boldsymbol{\xi}\text{)}$$
$$\mu := \mathbb{E}_{\boldsymbol{x},\boldsymbol{\xi}}f(\boldsymbol{x},\boldsymbol{\xi}) = \mathbb{E}_{\boldsymbol{\xi}}\mu_{\boldsymbol{\xi}} \quad \text{(global mean)}.$$

Expand the total variance

$$\text{Var}_{\boldsymbol{x},\boldsymbol{\xi}}[f(\boldsymbol{x},\boldsymbol{\xi})] = \mathbb{E}_{\boldsymbol{x},\boldsymbol{\xi}}\left[\|f(\boldsymbol{x},\boldsymbol{\xi}) - \mu\|^2\right] = \mathbb{E}_{\boldsymbol{\xi}}\left[\mathbb{E}_{\boldsymbol{x}}\left[\|f(\boldsymbol{x},\boldsymbol{\xi}) - \mu\|^2\right]\right].$$

Now insert and subtract $\mu_{\boldsymbol{\xi}}$ inside the norm:

$$\|f(\boldsymbol{x},\boldsymbol{\xi}) - \mu\|^2 = \|f(\boldsymbol{x},\boldsymbol{\xi}) - \mu_{\boldsymbol{\xi}} + \mu_{\boldsymbol{\xi}} - \mu\|^2$$
$$= \|f(\boldsymbol{x},\boldsymbol{\xi}) - \mu_{\boldsymbol{\xi}}\|^2 + 2\langle f(\boldsymbol{x},\boldsymbol{\xi}) - \mu_{\boldsymbol{\xi}}, \mu_{\boldsymbol{\xi}} - \mu\rangle + \|\mu_{\boldsymbol{\xi}} - \mu\|^2.$$

Taking expectation over $\boldsymbol{x}$ (for fixed $\boldsymbol{\xi}$), the cross term vanishes:

$$\mathbb{E}_{\boldsymbol{x}}[f(\boldsymbol{x},\boldsymbol{\xi}) - \mu_{\boldsymbol{\xi}}] = 0 \quad \Rightarrow \quad \mathbb{E}_{\boldsymbol{x}}\left[\langle f(\boldsymbol{x},\boldsymbol{\xi}) - \mu_{\boldsymbol{\xi}}, \mu_{\boldsymbol{\xi}} - \mu\rangle\right] = 0.$$

Therefore

$$\mathbb{E}_{\boldsymbol{x}}\left[\|f(\boldsymbol{x},\boldsymbol{\xi}) - \mu\|^2\right] = \mathbb{E}_{\boldsymbol{x}}\left[\|f(\boldsymbol{x},\boldsymbol{\xi}) - \mu_{\boldsymbol{\xi}}\|^2\right] + \|\mu_{\boldsymbol{\xi}} - \mu\|^2.$$

Finally take expectation over $\boldsymbol{\xi}$:

$$\text{Var}_{\boldsymbol{x},\boldsymbol{\xi}}[f(\boldsymbol{x},\boldsymbol{\xi})] = \mathbb{E}_{\boldsymbol{\xi}}\left[\mathbb{E}_{\boldsymbol{x}}\left[\|f(\boldsymbol{x},\boldsymbol{\xi}) - \mu_{\boldsymbol{\xi}}\|^2\right]\right] + \mathbb{E}_{\boldsymbol{\xi}}\left[\|\mu_{\boldsymbol{\xi}} - \mu\|^2\right]$$
$$= \mathbb{E}_{\boldsymbol{\xi}}\left[\text{Var}_{\boldsymbol{x}}f(\boldsymbol{x}|\boldsymbol{\xi})\right] + \text{Var}_{\boldsymbol{\xi}}\left[\mathbb{E}_{\boldsymbol{x}}f(\boldsymbol{x}|\boldsymbol{\xi})\right].$$

This completes the proof of the decomposition. $\square$

Based on the decomposition, the first term $\mathbb{E}_{\boldsymbol{\xi}}\left[\text{Var}_{\boldsymbol{x}}f(\boldsymbol{x}|\boldsymbol{\xi})\right]$ represents the input variability, and the second term $\text{Var}_{\boldsymbol{\xi}}\left[\mathbb{E}_{\boldsymbol{x}}f(\boldsymbol{x}|\boldsymbol{\xi})\right]$ represents the internal variability. Treating the input perturbations as signals and internal randomness as noise, we can further define the SNR in eq. 7.

## B THE DISTRIBUTION OF MINIMUM PERTURBATIONS AT THE ANCHOR POINTS

In Sec. 5, we showed that the compression and robustness follow a sigmoidal relation arising from the distribution of minimum perturbations of the classifier at the clean samples. Precisely speaking, this argument only holds if the distribution of the minimum perturbations of the classifier at the clean points is similar to that at the anchor points. In this section, we verify that it is indeed the case, as long as the clean accuracy at the anchor point is close to that at the clean samples. Predicting at the anchor points, would, however, always provide a slightly better estimation.

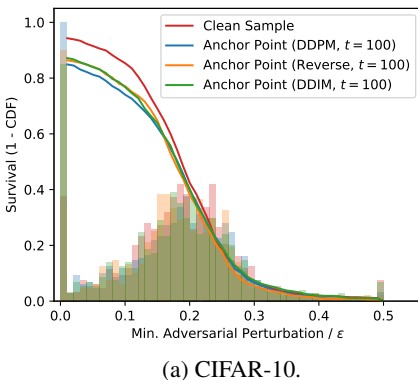 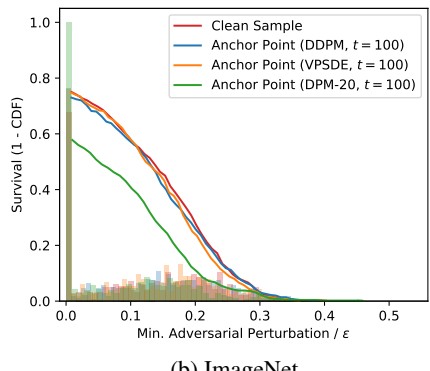

(a) CIFAR-10.      (b) ImageNet.

Figure S1: **The distribution of minimum perturbations at the anchor points and the survivial functions (1-CDF).** For both CIFAR-10 and ImageNet, and across the anchor points of discrete and continuous sampling methods, the minimum perturbation distribution all exhibit a zero-inflated Gaussian shape, where the zero-mass corresponding to the clean accruary at the anchor points.

We have measured the distributions of the classifier's minimum perturbations at the anchor points of DDPM, Reverse-only DDPM, and DDIM models on CIFAR-10, and at the anchor points of DDPM, VPSDE, and DPM-20 on ImageNet. As shown in Fig. S1, in all cases, the distributions exhibit a zero-inflated Gaussian shape. Consequently, their survival functions (i.e., $1 - \mathrm{CDF}$) take on a sigmoidal form. The y-intercept of the survival function, arising from the zero-mass component, corresponds precisely to the clean accuracy at the anchor point. As a result, the sigmoidal curves at the anchor points closely match those at the clean samples whenever the clean accuracies remain similar (e.g., for $t \leq 100$). This condition is necessary for any adversarial purification procedure to be meaningful. When $t$ becomes large and the clean accuracy at the anchor points drops substantially, the two sigmoidal curves naturally diverge. Therefore, in the regime where adversarial purification is meaningful—that is, where the clean accuracy at the anchor points does not significantly decrease relative to that of the original samples—the sigmoidal relation at the anchor points can be well approximated by that of the clean samples. This approximation greatly simplifies the analysis, as the resulting sigmoidal structure becomes an intrinsic property of the classifier itself, independent of the purification process. Incorporating the small deviations between the two distributions would further refine the quantitative precision of our theory.

## C  EVALUATING ROBUSTNESS OF DIFFUSION MODELS WITHOUT STOCHASTICITY

The results in Sec. 4 suggest that internal randomness in diffusion models plays a dominant role in determining the final purified output. This implies that randomness may significantly influence the empirical evaluation of adversarial robustness. In general, how to properly handle randomness in robustness evaluation has been debated (Athalye et al., 2018; Carlini et al., 2019; Gao et al., 2022; Yoon et al., 2021). Importantly, a concurrent work (Liu et al., 2025) shared the same view of evaluating the robustness of diffusion models with stochasticity fixed with empirical evaluations of EOT. We further argue that EOT without controlling randomness should be interpreted as a transfer attack. In the following section, we carefully examine how randomness affects the evaluation of diffusion-based purification and its implications for interpreting robustness gains.

**EOT as a transfer attack.** Following the notations in Sec. 2, let $L$ be the loss function and $t$ be the target associated with $\boldsymbol{x}_0$, the gradients during attack and the system during evaluation can be expressed as

$$\text{Attack: } \nabla_{\boldsymbol{x}} L\left[g(f_{\boldsymbol{\xi}_{\text{attack}}}(\boldsymbol{x})), t\right], \quad \text{Test: } g(f_{\boldsymbol{\xi}_{\text{test}}}(\boldsymbol{x})). \tag{21}$$

It is important to realize that the noise samples for calculating the attack $\boldsymbol{\xi}_{\text{attack}}$ and for the test $\boldsymbol{\xi}_{\text{test}}$ are typically different, although they are drawn from the same distribution. This means the attack is optimizing against a different function than the one will be used at test time. As a result, the attack becomes suboptimal and should be viewed as a transfer attack. This is particularly concerning in the context of diffusion models, where internal noise heavily influences the final output (as shown in the previous section). In this case, the discrepancy between $f_{\boldsymbol{\xi}_{\text{attack}}}$ and $f_{\boldsymbol{\xi}_{\text{test}}}$ can be substantial, limiting the attack's performance to evaluate the robustness of the purification system.

Expectation-over-transformation (EOT) (Athalye et al., 2018) was proposed to address the sub-optimality introduced by stochastic gradients:

$$\text{Attack (EOT): } \mathbb{E}_{\boldsymbol{\xi}} \nabla_{\boldsymbol{x}} L[g(f_{\boldsymbol{\xi}_{\text{attack}}}(\boldsymbol{x})), t] = \nabla_{\boldsymbol{x}} \mathbb{E}_{\boldsymbol{\xi}} L[g(f_{\boldsymbol{\xi}_{\text{attack}}}(\boldsymbol{x})), t], \quad \text{Test: } g(f_{\boldsymbol{\xi}_{\text{test}}}(\boldsymbol{x})). \tag{22}$$

Crucially, although EOT mitigates the effect of randomness in gradient computation by marginalizing over the noise during attack, randomness still remains at test time. PGD-EOT attack may be interpreted as approximatly calculating the gradient of the average ensemble system $\mathbb{E}_{\boldsymbol{\xi}} g(f_{\boldsymbol{\xi}_{\text{attack}}}(\boldsymbol{x}))$, and later applied to the non-ensemble system $g(f_{\boldsymbol{\xi}_{\text{test}}}(\boldsymbol{x}))$. [2] To this end, PGD-EOT can also be interpreted as a transfer attack. While EOT improves gradient quality compared to single-sample attacks (as confirmed empirically), the attack remains suboptimal. We quantify this by computing the correlation between PGD-EOT gradients and those from the optimal attack that directly targets $g(f_{\boldsymbol{\xi}_{\text{test}}}(\boldsymbol{x}))$. We find a mean correlation of $0.1682 \pm 0.0976$ (see Table S5), supporting the view that PGD-EOT does not fully capture the optimal gradients of the defense system.

Here we propose an alternative evaluation method by fixing the randomness during both attack generation and testing—specifically, setting $\boldsymbol{\xi}_{\text{attack}} = \boldsymbol{\xi}_{\text{test}}$. This approach eliminates the mismatch in stochasticity between the attack and defense, thereby addressing concerns related to suboptimal gradients and transfer attacks. It enables a more faithful estimation of the robustness of adversarial purification systems independent of randomness. As we will demonstrate below, even under fully controlled randomness, diffusion models still exhibit non-trivial robustness, although the observed robust accuracy is significantly lower than previous reports.

**Robustness of diffusion models without stochasticity.** Prior work (Nie et al., 2022) reported that, when using a diffusion timestep of t=100, the empirically reported robustness on CIFAR-10 using PGD-EOT is about 70%. We control the randomness within diffusion models by controlling the random seeds during both the forward and reverse processes (see Appendix D for details of the implementation). After controlling the effect of randomness, we find that the robustness gain of diffusion models is 23.7% on CIFAR-10 (PGD). For ImageNet, when fixing the randomness, the BPDA attack has a robustness of 29.5%. These numbers are substantially lower than those reported previously (also see Tables 2 & 3).

---

[2]For this approximation to be exact, the expectation operation needs to commute with the loss function $L$. This in general does not hold for cross-entropy loss, but was not explicitly discussed in Athalye et al. (2018).

Table S2: Robustness of diffusion models w/o. stochasticity on CIFAR-10 ($\ell_\infty = 8/255$, $t = 100$).

| Model | Fix Random | Clean Acc. | PGD | PGD-EOT |
|-------|:---:|:---:|:---:|:---:|
| DDPM | ✗ | 86.0±0.8% | 71.9±0.2% | 59.3% |
|       | ✓ | 85.8±0.4% | **23.7±0.7%** | – |

Table S3: Robustness of diffusion models w/o. stochasticity on ImageNet ($\ell_\infty = 4/255$, $t = 150$).

| Model | Fix Random | Clean Acc. | BPDA | BPDA-EOT |
|-------|:---:|:---:|:---:|:---:|
| Guided | ✗ | 67.2±2.4% | 63.7±1.2% | 59.0% |
|        | ✓ | 68.5±0.8% | **29.5±0.4%** | – |

Table S4: Transfer attack across different random configurations.

| Dataset | Fix Random | PGD/BPDA |
|---------|:---:|:---:|
| CIFAR-10 | ✓ | 77.4±0.36% |
| ImageNet | ✓ | 66.3±2.32% |

Table S5: Correlations between PGD/EOT attacks on CIFAR-10.

| Attacks | Correlation |
|---------|:---:|
| PGD (Fix) vs. PGD (Fix) | 0.0818±0.0709 |
| PGD-EOT vs. PGD (Fix) | 0.1682±0.0976 |

**Transfer attack across random configurations.** We further conduct transfer attack experiments to illustrate how randomness effect robustness evaluation. We calculated the attack with a fixed randomness configuration, and evaluated with a different fixed randomness configuration. Based on our theoretical reasoning above, we hypothesize that should mimic the effect of not controlling randomness. In support of our hypothesis, the observed empirical robustness accuracies for both CIFAR-10 and ImageNet are generally consistent with those reported using reported previously (Table S4). Additionally, we measure the correlation between the adversarial perturbations generated under different random configurations and find a mean correlation of 0.0818±0.0709 (Table S5).

## D  IMPLEMENTATION DETAILS OF ADVERSARIAL ATTACKS ON DIFFUSION MODELS

**Datasets and base classifiers**  The experiments were conducted on the CIFAR-10 (Krizhevsky & Hinton, 2009) and ImageNet (Deng et al., 2009) datasets. For CIFAR-10, we subsampled the first 1000 images from the test set. For ImageNet, we subsampled the first 200 images from the validation set. Standard preprocessing was applied to the datasets. We used the standard classifiers from the `RobustBench` (Croce et al., 2020)`https://github.com/RobustBench/robustbench`. Namely, the WideResNet-28-10 model for CIFAR-10, and ResNet-50 model for ImageNet. The clean accuracy on our subsampled set for the classifier is 94.8% on CIFAR-10 (vs. 94.78% on the full set) and 74.5% on ImageNet (vs. 76.52% on the full set).

**Diffusion models**  We focused on discrete-time diffusion models in this paper to avoid the potential gradient masking induced by numerical solvers in continuous-time models (Huang et al., 2022). For CIFAR-10, we used the official checkpoint of DDPM (coverted to PyTorch from Tensorflow `https://github.com/pesser/pytorch_diffusion`) instead of Score-SDE. For ImageNet, we used the official checkpoint of $256 \times 256$ unconditional Guided diffusion (Dhariwal & Nichol, 2021) `https://github.com/openai/guided-diffusion` as the purification system. The purification time steps were kept the same with Nie et al. (2022), namely $t^* = 0.1$ (100 forward and 100 reverse steps) for CIFAR-10 and $t^* = 0.15$ (150 forward and 150 reverse steps) for ImageNet.

The DiffPure (Nie et al., 2022) framework proposed to utilize both the forward and reverse processes of diffusion models for adversarial purification. Since the forward process introduces a large amount of randomness, we explore whether it's possible to remove the forward process, thus only using the reverse process of diffusion models for adversarial purification. A similar reverse-only framework was proposed in DensePure (Xiao et al., 2023), but further equipped with a majority voting mechanism to study the certificated robustness.

**Fixing randomness in diffusion models**  We controlled the randomness within diffusion models by controlling the random seeds during both the forward and reverse processes. For the base seed $s$, $i$-th batch of data at the $t$ step of the forward/reverse process, we set the random seed

$$\texttt{seed}(s, i, t) = \begin{cases} \texttt{hash}(s, i, 2t), & \text{if forward process} \\ \texttt{hash}(s, i, 2t + 1), & \text{if reverse process} \end{cases} \tag{23}$$

before sampling the Gaussian noise from eq. 3 or eq. 4. The multiplicative hashing function

$$\texttt{hash}(s, i, t) = (p_s \cdot s) \oplus (p_i \cdot i) \oplus (p_t \cdot t) \mod 2^{32} \tag{24}$$

where $p_s, p_i, p_t$ are large numbers coprime with each other to avoid collision and $\oplus$ denotes bitwise XOR. This setting ensures that we have a different random seed for each batch of data and timesteps in the forward/reverse process, but will keep the randomness the same through the entire purification process if encountering the same data batch.

**Adversarial attacks**  We conducted BPDA/BPDA-EOT and PGD/PGD-EOT attacks (Athalye et al., 2018) on CIFAR-10 with $\ell_\infty = 8/255$, and BPDA/BPDA-EOT attacks on ImageNet with $\ell_\infty = 4/255$. The PGD was conducted based on the `foolbox` (Rauber et al., 2020)`https://github.com/bethgelab/foolbox`, and the BPDA wrapper was adapted from `advertorch` (Ding et al., 2019)`https://github.com/BorealisAI/advertorch`. Full gradients were calculated for the PGD/PGD-EOT as (Lee & Kim, 2023) discovered that the approximations methods used in the original DiffPure (Nie et al., 2022) incurred weaker attacks. The full gradient of PGD/PGD-EOT is the strongest attack for DiffPure methods according to Lee & Kim (2023) experiments, and is very computationally expensive. We ran our CIFAR-10 attack experiments on a NVIDIA RTX 6000 GPU for 10 days. We were not able to conduct the full PGD attack on ImageNet in a reasonable time given our available resources. The key hyperparameters for our attacks are listed in Table S6. All attacks were repeated three times (n = 3) to compute the standard deviation, except for the EOT experiments, for which we were unable to do so due to limited computational resources. We used the base seeds $s = 0, 1, 2$ for all experiments. We use additive uniform noise $U[-\epsilon, \epsilon]$ to estimate the expected CR, as it allows convenient control of the $\ell_\infty$ norm. Strictly speaking, uniform noise is not isotropic; however, we find empirically that the difference is negligible when $\epsilon$ is small.

**FID score** The FID score was calculated based on the `pytorch-fid` package (Seitzer, 2020)`https://github.com/mseitzer/411pytorch-fid`. We compute the FID score using 768-dimensional pre-classifier features due to the limited number of PGD samples .

Table S6: Hyperparameters for adversarial attacks.

| Hyperparameters | Values {CIFAR-10, ImageNet} |
|---|---|
| Attack magnitude | {8, 4} / 255 |
| PGD steps | 40 |
| Relative PGD step size | 0.01 / 0.3 |
| EOT numbers | 15 |
| Batch size | 1 |
| Random factor $p_s$ | 83492791 |
| Random factor $p_i$ | 73856093 |
| Random factor $p_t$ | 19349663 |

# E    ADDITIONAL EXPERIMENTAL RESULTS

## E.1    $\ell_p$ DISTANCE MEASUREMENTS DURING DIFFPURE

Additional distance measurements during the DiffPure process are shown in Fig. S2 and S3. For CIFAR-10, we further measured the $\ell_\infty$ distances for the experiment illustrated in Fig. 1c. For ImageNet, we repeated the same experiment with the unconditional Guided diffusion with 150 diffusion and denoising steps ($t^* = 0.15$, the same setting with the DiffPure (Nie et al., 2022)), and measured the $\ell_2/\ell_\infty$ distances. The distances during the intermediate diffusion process in ImageNet (Fig. S3) are not shown as the code base implemented the one-step diffusion equation equivalent to the multistep diffusion. Again, similar effects were observed under both $\ell_2/\ell_\infty$ distances across datasets, namely, diffusion models purified to states further away from the clean images, considerably larger than the original adversarial perturbation ball. Detailed data points are listed in Table S8,S9,S7. Specifically, the $\ell_2/\ell_\infty$ distances to clean samples at the init point ($t = 0$, the scale of the original perturbation), maximum point ($t = 100/150$, after forward diffusion), and end point ($t = 200/300$, after the reverse denoising). The end point distances are roughly 4 or 5 times of the size of the adversarial ball under $\ell_2$ distance on CIFAR-10/ImageNet, and 10 or 26 times under $\ell_\infty$ distance. Diffusion models transit back to the $\ell_2$ shrinkage regime beyond the uniform noise of $\epsilon = 16/255$, which is twice of the standard $\ell_\infty$ adversarial ball considered for CIFAR-10.

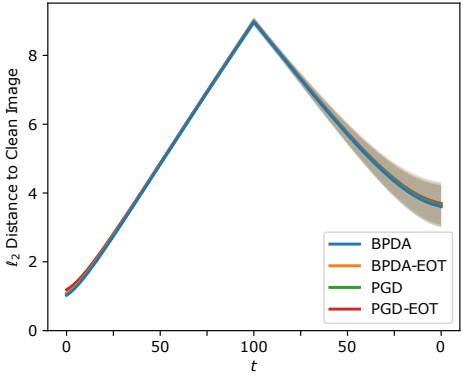
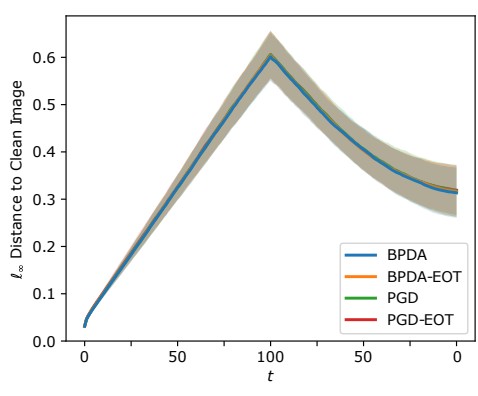

(a) $\ell_2$ distances, CIFAR-10.                    (b) $\ell_\infty$ distances, CIFAR-10.

Figure S2: Additional distance measurements during DiffPure on CIFAR-10.

## E.2    BEHAVIOR OF DIFFUSION MODELS UNDER RANDOM PERTURBATIONS

We wonder if the behavior of adversarial attacks under diffusion models is special at all, that is, whether the push-away phenomena we observed are in fact general to arbitrary perturbations around the clean images. To test this, we generated perturbations of clean images by sampling random noise

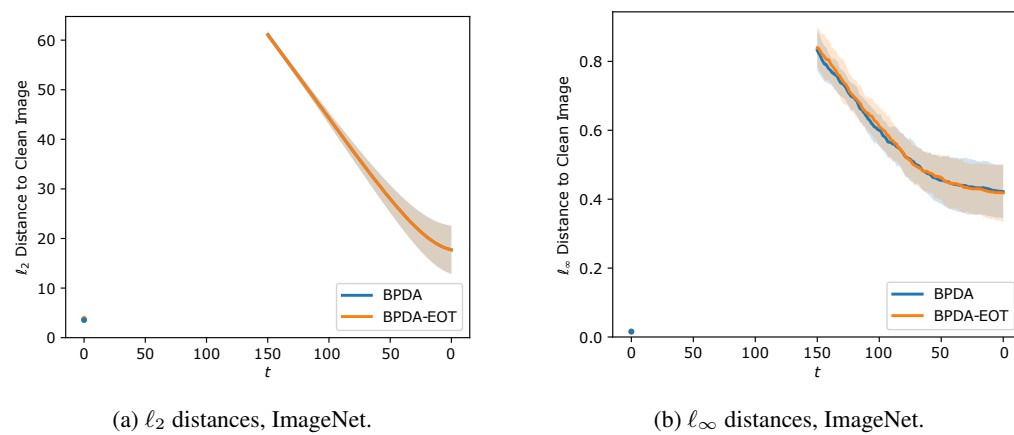

(a) $\ell_2$ distances, ImageNet.

(b) $\ell_\infty$ distances, ImageNet.

Figure S3: Additional distance measurements during DiffPure on ImageNet.

Table S7: $\ell_2/\ell_\infty$ distances measurements during DiffPure on ImageNet ($\ell_\infty = 4/255$).

| Distances | Attack | Init ($t = 0$) | Max ($t = 150$) | End ($t = 300$) |
|---|---|---|---|---|
| $\ell_2$ | BPDA | $3.537 \pm 0.079$ | $61.116 \pm 0.738$ | $17.712 \pm 4.851$ |
| | BPDA-EOT | $3.772 \pm 0.139$ | $61.078 \pm 0.762$ | $17.694 \pm 4.838$ |
| $\ell_\infty$ | BPDA | $0.016 \pm 0.000$ | $0.832 \pm 0.059$ | $0.422 \pm 0.077$ |
| | BPDA-EOT | $0.016 \pm 0.000$ | $0.839 \pm 0.060$ | $0.418 \pm 0.084$ |

uniformly with a fixed magnitude. We first tested small perturbations that match the size of the adversarial attack on CIFAR-10 ($\ell_\infty = 8/255$ uniform noise). We found that the behavior of the model under random noise (Fig. S4a, blue curve) is almost identical to that induced by adversarial attack (Fig. 1c, blue curve). These results, together with those reported above, suggest that diffusion models are not able to reduce the distances to a clean image from a slightly perturbed clean image. This raised the intriguing possibility that the clean images do not reside on the local peaks of the image priors learned in the diffusion models. This may make sense given the in memorization v.s. generalization trade-off (Kadkhodaie et al., 2024). That is, a model simply encodes every clean image as the prior mode may not generalize well.

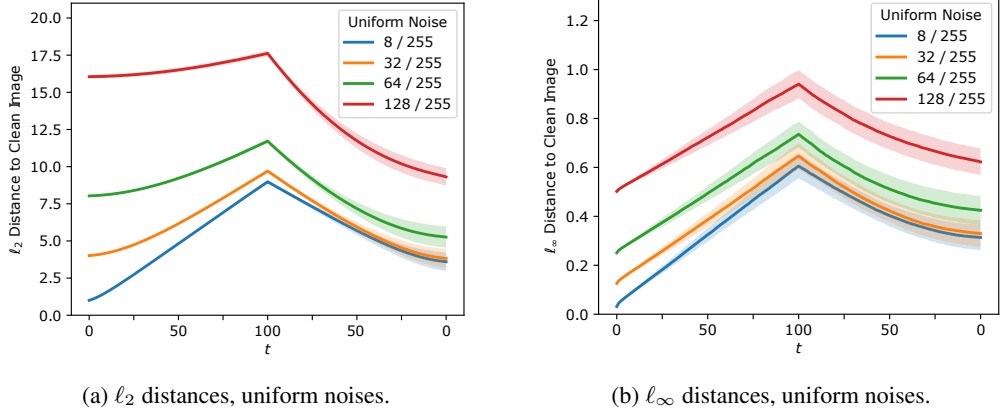

(a) $\ell_2$ distances, uniform noises.

(b) $\ell_\infty$ distances, uniform noises.

Figure S4: Distance measurements under random perturbations on CIFAR-10.

Although the results above indicate that diffusion models are ineffective in removing small perturbations, it is possible that they may be more effective in removing noise induced by larger perturbations. We performed the same $\ell_2$ distance analysis using three larger levels of uniform noises, ranging from $\epsilon = \{32, 64, 128\}/255$, to examine the model behavior under larger perturbations. As the noise level

Table S8: $\ell_2$ distance measurements during DiffPure on CIFAR-10 ($\ell_\infty = 8/255$).

| Attack | Init ($t = 0$) | Max ($t = 100$) | End ($t = 200$) |
|---|---|---|---|
| BPDA | $1.027 \pm 0.023$ | $8.976 \pm 0.118$ | $3.606 \pm 0.607$ |
| BPDA-EOT | $1.072 \pm 0.046$ | $8.992 \pm 0.116$ | $3.607 \pm 0.615$ |
| PGD (Full) | $1.077 \pm 0.040$ | $8.980 \pm 0.118$ | $3.646 \pm 0.614$ |
| PGD-EOT | $1.188 \pm 0.072$ | $8.999 \pm 0.115$ | $3.695 \pm 0.617$ |
| Uniform ($\epsilon = 8/255$) | $1.004 \pm 0.009$ | $8.979 \pm 0.113$ | $3.598 \pm 0.618$ |
| Uniform ($\epsilon = 16/255$) | $4.015 \pm 0.034$ | $9.699 \pm 0.124$ | $3.823 \pm 0.640$ |
| Uniform ($\epsilon = 32/255$) | $8.030 \pm 0.065$ | $11.715 \pm 0.145$ | $5.258 \pm 0.714$ |
| Uniform ($\epsilon = 128/255$) | $16.051 \pm 0.129$ | $17.622 \pm 0.184$ | $9.307 \pm 0.581$ |

Table S9: $\ell_\infty$ distance measurements during DiffPure on CIFAR-10 ($\ell_\infty = 8/255$).

| Attack | Init ($t = 0$) | Max ($t = 100$) | End ($t = 200$) |
|---|---|---|---|
| BPDA | $0.031 \pm 0.000$ | $0.601 \pm 0.051$ | $0.313 \pm 0.053$ |
| BPDA-EOT | $0.031 \pm 0.000$ | $0.603 \pm 0.051$ | $0.316 \pm 0.053$ |
| PGD (Full) | $0.031 \pm 0.000$ | $0.606 \pm 0.050$ | $0.317 \pm 0.055$ |
| PGD-EOT | $0.031 \pm 0.000$ | $0.606 \pm 0.050$ | $0.319 \pm 0.053$ |
| Uniform ($\epsilon = 8/255$) | $0.031 \pm 0.000$ | $0.605 \pm 0.050$ | $0.314 \pm 0.052$ |
| Uniform ($\epsilon = 16/255$) | $0.125 \pm 0.000$ | $0.647 \pm 0.054$ | $0.330 \pm 0.055$ |
| Uniform ($\epsilon = 32/255$) | $0.251 \pm 0.000$ | $0.735 \pm 0.052$ | $0.425 \pm 0.059$ |
| Uniform ($\epsilon = 128/255$) | $0.502 \pm 0.000$ | $0.941 \pm 0.057$ | $0.623 \pm 0.054$ |

increases, the $\ell_2$ distances of the final purified states increase. Interestingly, the model transits from "pushing-away" to "shrinkage" under very large perturbations.

### E.3 DISTRIBUTIONAL AND SEMANTIC DISTANCES FAIL TO EXPLAIN ROBUSTNESS IMPROVEMENTS

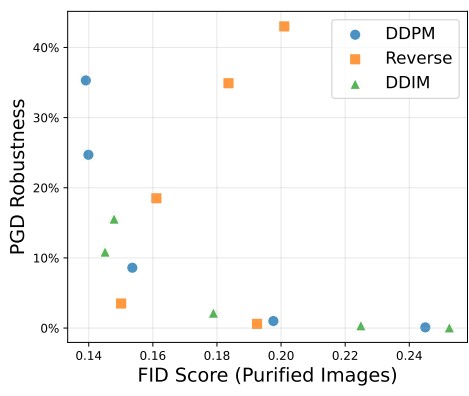 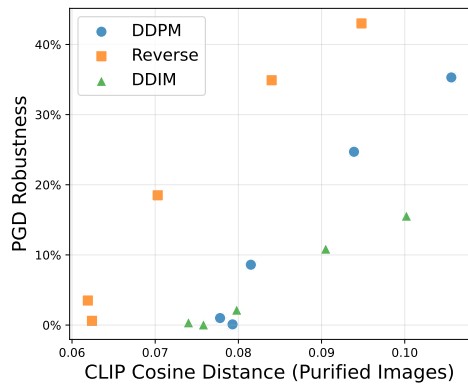

(a) FID distance after purification.        (b) Cosine distance in the CLIP space.

Figure S5: Distributional and semantic distances to the clean samples and adversarial robustness. For both metrics, we do not observe a monotonic relation with robustness, in contrast to our proposed compression rate, which displays a consistent monotonic relation (Fig. 3a).

We further measured the cosine distance between the adversarial samples and their corresponding clean samples in the CLIP representation space, both before and after purification. As shown in Table S11, we observed a reduction in distance for smaller timesteps $t$, whereas for larger $t$ the distance increases.

Similarly, if we treat the purified CLIP distance to the clean samples as a robustness indicator, the results are shown in Fig. S5b. Overall, neither FID nor CLIP semantic distances exhibit a consistent monotonic relationship with robustness, indicating that they are not reliable robustness indicators.

Table S10: FID to clean samples after purification

| Method | $t = 10$ | $t = 20$ | $t = 50$ | $t = 100$ | $t = 150$ |
|--------|----------|----------|----------|-----------|-----------|
| DDPM | 0.245 | 0.198 | 0.154 | 0.140 | 0.139 |
| Reverse | 0.192 | 0.150 | 0.161 | 0.184 | 0.201 |
| DDIM | 0.253 | 0.225 | 0.179 | 0.145 | 0.148 |

Table S11: CLIP distance reduction after purification.

| Method | $t = 10$ | $t = 20$ | $t = 50$ | $t = 100$ | $t = 150$ |
|--------|----------|----------|----------|-----------|-----------|
| DDPM | 0.0285 | 0.0249 | 0.0082 | -0.0162 | -0.0330 |
| Reverse | 0.0456 | 0.0359 | 0.0100 | -0.0135 | -0.0272 |
| DDIM | 0.0327 | 0.0333 | 0.0195 | 0.0009 | -0.0131 |

This contrasts with the strong and consistent correlation observed with our proposed expected compression rate, further supporting our claim that robustness arises from a global compression of the image space rather than proximity to clean samples in FID or semantic space.

### E.4 VARIANCES DECOMPOSITION AND SNR MEASUREMENTS

To quantify the relative contributions of input v.s. internal variability, we randomly selected 50 images from the CIFAR-10 test set. For each image, we generated 50 uniform noise perturbations within a fixed $\ell_\infty = 8/255$ norm bound and passed each perturbed input through the diffusion model using 50 different random seeds. This resulted in a total of 125,000 purification images. We then estimated the variance components and SNR as defined. The results are summarized in Table S12 and S13.

Table S12: Variances and SNRs in diffusion purification $/\times 10^{-3}$.

| Total Var. | Input Var. | Internal Var. | SNR |
|------------|------------|---------------|-----|
| 3.597±0.877 | 0.020±0.005 | 3.578±0.873 | 5.926±1.065 |

Table S13: $\ell_2$ Distances to corresponding centroids in diffusion purification.

| Initial | Fix Inputs | Fix Random | Vary Both |
|---------|------------|------------|-----------|
| 1.004±0.001 | 0.241±0.032 | 3.282±0.453 | 3.288±0.453 |

### E.5 THE EFFECT OF TIMESTEPS ON CLEAN ACCURACY AND COMPRESSION RATES

We investigated the effect of timesteps on compression rate, robustness and clean accuracy. As timesteps increases, the clean accuracy drops monotonically, as there are more samples mis-purified into different classes. Intriguingly, for reverse-only diffusion models, we observed a S-shape trend on the compression rates, which can be divided into three regimes (Fig. S6b). In low timesteps (0-200), compression rates decreases as timesteps increases. In medium timesteps (200-700), there is a increase of compression rates, which constraints the robustness improvements. Lastly in the high timesteps (700-1000), the compression rate further reduces, but due to the low clean accuray, the robustness improvement is also limited. However, we did not observe the same trend for DiffPure methods, as the compression rate is monotonically decreasing (Fig. S6a).

## F COMPUTATIONAL RESOURCES AND REPRODUCIBILITY

We conduct our PGD/BPDA experiments on Nvidia GeForce 3080Ti GPU. The PGD/BPDA experiments on CIFAR-10/ImageNet took around 24 hours for each repeat with a batchsize of 1 on our subsampled dataset. For PGD/BPDA-EOT experiments, we rent Nvidia H100 GPU (80GB), and the experiments took around 120 hours for batchsize of 1. The codebase will be open-sourced in the camera-ready version once published.

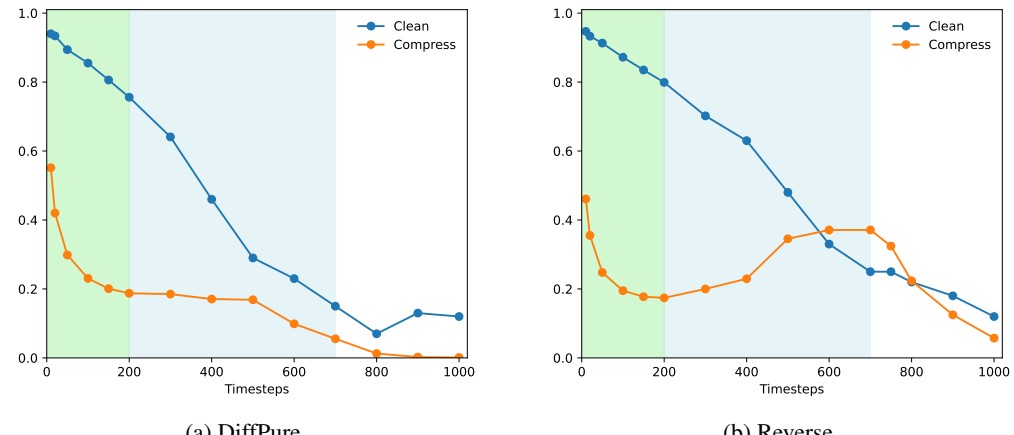

(a) DiffPure.               (b) Reverse.

Figure S6: The effect of timesteps on clean accuracy and compression rates.

## G  THE USE OF LARGE LANGUAGE MODELS (LLMS)

Large Language Models (LLMs) were used primarily for text polishing. No paragraphs were originally written by LLMs. LLMs also assisted in clarifying mathematical derivations and generating functional code from instructions. No figures were generated by LLMs. The authors take full responsibility for all content presented in this paper. LLMs are not eligible for authorship.

## H EXAMPLES OF ADVERSARIAL AND PURIFIED IMAGES WITH DIFFERENT COMPRESSION RATES

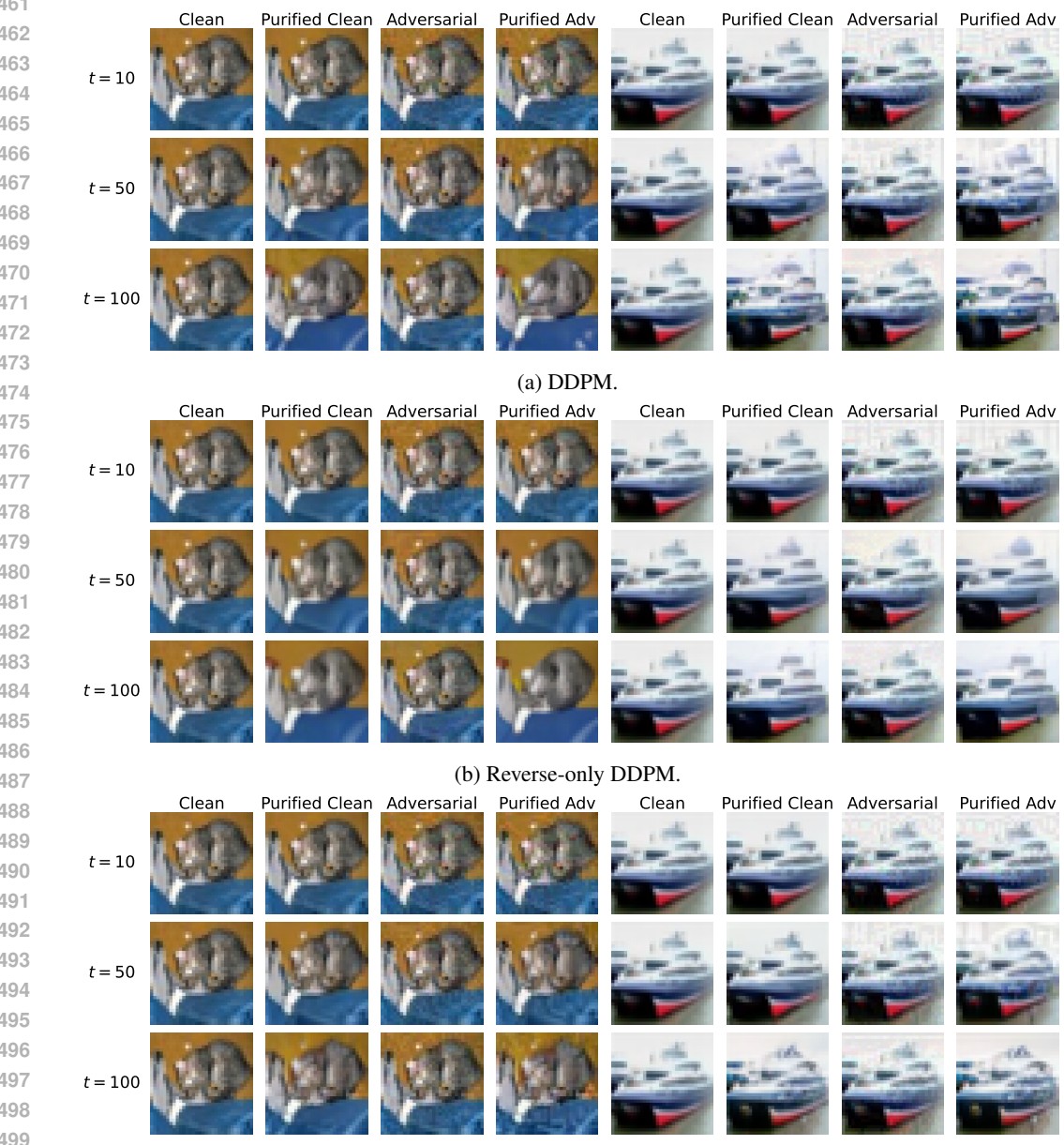

(a) DDPM.

(b) Reverse-only DDPM.

(c) DDIM.

Figure S7: Examples of adversarial and purified images on CIFAR-10.

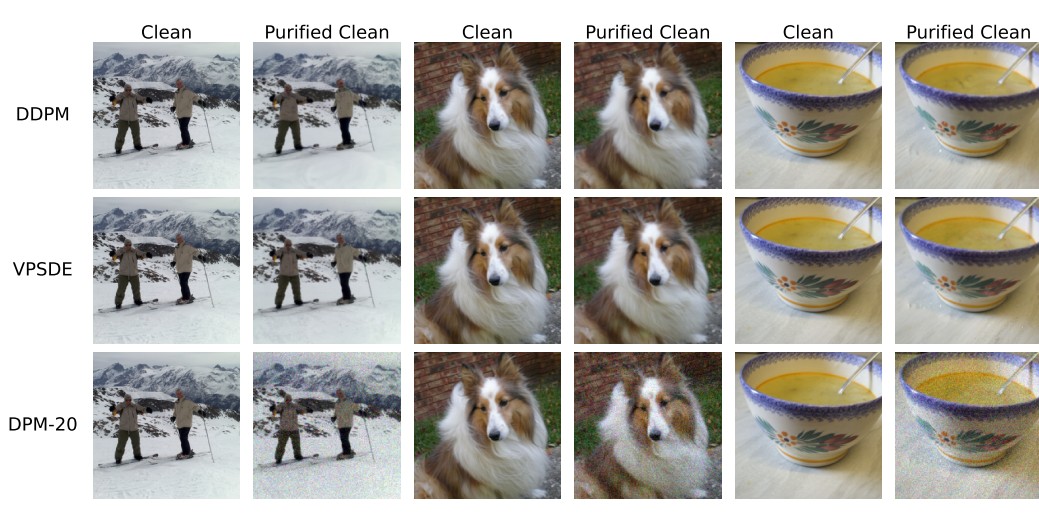

(a) $t = 100$.

Figure S8: Examples of adversarial and purified images on ImageNet.

