# OpenReview forum: "Diffusion Models Improve Adversarial Robustness by Compressing Image Space"
_ICLR.cc/2026/Conference — Submitted to ICLR 2026_

### Official Review · Reviewer_xbFW · 2025-10-29

**Soundness:** 2
**Presentation:** 2
**Contribution:** 3
**Rating:** 6
**Confidence:** 4

**Summary:**

This paper investigates how diffusion-based purification methods improve adversarial robustness. First, a counterintuitive phenomenon is observed that diffusion-based purification increases the distance to clean samples rather than decreases it, shifting the perturbed samples around the clean image to a new space around an anchor point. Then, by disentangling the variability in the input perturbation and purification stochasticity, it is identified that the image space is compressed when the system's random states are fixed. Further empirical and theoretical analyses attribute the improved robustness under fixed random states to the compression effect, especially along off-manifold directions.

**Strengths:**

1. This paper provides new insights into the mechanism of diffusion-based adversarial purification methods, including the counterintuitive finding of increased image distance after purification, and the disentanglement of the contribution of stochasticity and image space compression. These insights may also motivate other research on diffusion models.
1. The viewpoint that EOT should be understood as a transfer attack is interesting and reasonable.
1. The main ideas of the paper are clearly illustrated by the figures and supported by both empirical and theoretical analyses.

**Weaknesses:**

1. My major concern is the **relation between compression rates and adversarial robustness**. Although Fig. 3a empirically suggests the monotonic relation between robustness and compression rate, and this relation can potentially be explained by the distribution of minimal adversarial perturbation plotted in Fig. 3b, it cannot be concluded that compression contributes to the non-trivial robustness under fixed randomness. The key issue is that **the compressed space is around the shifted anchor points $f(x_0)$, while Fig. 3b is plotted based on the clean images $x_0$** according to Lines 369-371. If the distribution of minimal adversarial perturbation for anchor points is statistically similar to that of the clean images, then compressing the space can indeed increase the robustness.
2. The clarity of the figures and tables should be improved:
- The notation "Reverse" in Fig. 1, Table 1, and others is not clearly defined.
- In Fig. 2f, the bars for "fixing input" are almost completely covered by those for "varying both", which could lead to confusion.
- The details for fitting the robustness-compression curve in Fig. 3 should be clarified.
- The "PGD (Robust)" and "PGD (Non-robust)" in Table 2 are not explained and are confusing. If these refer to the compression rates along some adversarial directions, then it seems to contradict with the conclusions in Sec. 6.
- The notations "Compress" and "PGD (Fix)" in Table 3 should also be better clarified.
3. It would be better to provide more qualitative examples of clean and purified images to support the arguments in Sec. 3.
4. Table 1 is weak evidence for the non-monotonic relation between distributional distances and adversarial robustness, as it only incorporates three data points.
5. The timestep values for the reverse process in Sec. D (e.g., 100-200) are inconsistent with those in the main text (e.g., 100-0).

**Questions:**

1. The expected compression rate is computed based on isotropic noise. However, adversarial noises are not likely to be isotropic. Would this issue affect the analyses and conclusions?
1. While designing a compression-based purification system can be a promising direction, the existing results for diffusion-based methods (e.g., those in Sec. B) suggest that stochasticity may be more effective for adversarial defense. Is there any specific evidence or rationale that supports the potential superiority of compression-based methods?

---

> ### Author Response · Authors · 2025-11-21
>
> Thank you for your constructive and thoughtful feedback. Please find our responses to the identified weaknesses and questions below.
>
> # Weakness 1: Distribution of minimum perturbation at the anchor points
>
> Thank you for this excellent question. The short answer is yes—the distributions of minimum perturbations at the anchor points are **similar to those at the clean samples**, within the range of robustness levels that are of primary interest. We had these results during the preparation phase of the initial version of the manuscript, but chose not to describe these technical details in order to keep the main results focused, as the manuscript was already pretty dense. We appreciate the reviewer highlighting this aspect, as it gives us the opportunity to further discuss these details more explicitly.
>
> We have measured the distributions of the classifier’s minimum perturbations at the anchor points of DDPM, Reverse-only DDPM, and DDIM models across multiple timesteps $t$. In all cases, the distributions exhibit a zero-inflated Gaussian shape. Consequently, their survival functions (i.e., $1-\mathrm{CDF}$) take on a sigmoidal form. The y-intercept of the survival function, arising from the zero-mass component, **corresponds precisely to the clean accuracy at the anchor point**.
> As a result, the sigmoidal curves at the anchor points closely match those at the clean samples whenever the clean accuracies remain similar (e.g., for $t \leq 150$). This condition is necessary for any adversarial purification procedure to be meaningful. When T becomes large and the clean accuracy at the anchor points drops substantially, the two sigmoidal curves naturally diverge.
> Therefore, in the regime where adversarial purification is meaningful—that is, where the clean accuracy at the anchor points does not significantly decrease relative to that of the original samples—the sigmoidal relation at the anchor points can be well approximated by that of the clean samples. This approximation greatly simplifies the analysis, as the resulting sigmoidal structure becomes an intrinsic property of the classifier itself, independent of the purification process. Incorporating the small deviations between the two distributions would only refine the quantitative precision of our theory without altering its conclusions.
> We will include these results in the revised version.
>
> # Weakness 2: Improve the clarity of the figures and tables
> We thank the reviewer for listing the notations needed to be further clarified and will incorporate these in the final version. Here are our quick clarifications:
>
> * “Reverse” refers to the DDPM diffusion model with only the reverse process (Reverse-only).
> * Yes, the main point here is, the distributions of “fixing input” and “vary both” are almost exactly the same, showing that the internal randomness, rather than any signal from the input itself, dominates the behavior (extremely low SNR ratio).
> * The curve was fitted by a logistic function $L / (1 + exp(-k * (x-x_0)))$ on the compression rate and robustness data, with $L$, $k$, and $x_0$ as the free parameters.
> * These refer to the compression rate measured by the robust/non-robust samples of the PGD perturbations. This is another important question and please refer to our answer to Question 1.
> * Compress refers to the compression rate over isotropic noise, and PGD (Fix) refers to the PGD robustness with fixed randomness.
>
> # Weakness 3: Additional examples of clean and purified images
> More qualitative examples of clean and purified images will be provided in the appendix of the revised version.

---

> ### Author Response · Authors · 2025-11-21
>
> # Weakness 4: Additional evidence of non-monotonic relation between FID and robustness
> Thank you for this helpful suggestion. We measured the FID distance between the purified adversarial samples and the clean samples across multiple purification methods (DDPM, Reverse-only DDPM, and DDIM) and multiple timesteps ($t = 10, 20, 50, 100, 150$), yielding the 15 data points shown below:
>
> ### Table: FID to Clean Samples After Purification and Robustness
>
> | **Method** | **Metric** | **t=10** | **t=20** | **t=50** | **t=100** | **t=150** |
> |-----------|------------|----------|----------|----------|-----------|-----------|
> | **DDPM**  | FID        | 0.245    | 0.198    | 0.154    | 0.140     | 0.139     |
> |           | PGD  | 0.001    | 0.010    | 0.086    | 0.247     | 0.353     |
> | **Reverse** | FID      | 0.192    | 0.150    | 0.161    | 0.184     | 0.201     |
> |             | PGD  | 0.006    | 0.035    | 0.185    | 0.349     | 0.430     |
> | **DDIM**  | FID        | 0.253    | 0.225    | 0.179    | 0.145     | 0.148     |
> |           | PGD   | 0.001    | 0.004    | 0.022    | 0.109     | 0.156     |
>
> Plotting FID against robustness does not reveal a monotonic or predictive relationship, in contrast to the strong and consistent correlation observed with our proposed expected compression rate. This further supports our claim that robustness arises from global compression of the image space, rather than proximity to clean samples in FID.
>
>
>
> # Weakness 5: Inconsistent timestep labeling in the appendix
> We will fix this typo in the revised version and thanks for pointing it out.
>
> # Question 1: Expected CR over isotropic noises vs. CR of gradient-based attacks
> Thank you for this excellent question. The key finding of our paper is that the robustness achieved by diffusion-based adversarial purification is determined by the model’s ability to **compress the entire image space**, rather than its effect on individual samples. This insight motivates our title, “Compressing the Image Space.”
>
> We quantify this global compression using the expected compression ratio (CR) computed over isotropic noise directions. This expectation reflects the aggregate robustness across all perturbations, not just those aligned with specific adversarial gradients. This is consistent with the empirical behavior of adversarial purification systems, which often improve broad robustness, in contrast to adversarial training methods that typically enhance robustness only along particular adversarial directions and sometimes harm out-of-distribution robustness (e.g., Gaussian noise).
>
> Interestingly, although the compressed PGD perturbations show a consistent trend (e.g., robust samples tend to have smaller compressed perturbation lengths), the CR derived solely from gradient-based attacks does not reliably predict robustness. This indicates that robustness depends on more than just the magnitude of compressed adversarial perturbations—the direction also matters. In contrast, the expected CR, which captures the overall compression of the image space, exhibits a much clearer and more reliable relationship with robustness.
>
> This distinction supports our core observation: robustness in adversarial purification arises from the compression of the image space as a whole, not from properties of individual samples. To the best of our knowledge, this is a new insight that was not recognized in the prior literature.

---

> ### Author Response · Authors · 2025-11-21
>
> # Question 2: Why randomness would not be considered as a more promising approach for robustness improvement over compression?
> Our central argument is that the evaluated empirical robustness of purification systems can be **decomposed into two components**: (1) the part that arises from intrinsic properties of the system when randomness is fixed, and (2) the part that results solely from stochasticity during the attack and defense phases. For diffusion-based purification, we show that the first component can be explained through our proposed compression theory. Importantly, this decomposition applies not only to diffusion models but to all adversarial purification systems involving randomness.
> As analyzed in Section 4 and Appendix B, randomness appears to “improve” robustness because it breaks the consistency between the gradients used to generate adversarial examples and the gradients seen during defense. In other words, the attack and defense use **different noise realizations**, making the **attack gradients suboptimal or even meaningless**. This may be interpreted as  an artifact of evaluation, not a true indication of robustness.
> Based on our analysis, the effect of different randomness realizations can be interpreted as a form of transfer attack. As shown in Table S4, the robustness without controlling for randomness resembles the behavior observed when adversarial examples are transferred across different randomness configurations. In this setting, the computed adversarial directions are only weakly correlated with the true optimal direction. Moreover, as demonstrated both analytically and empirically in Table S5, Expectation over Transformation (EOT) fails to resolve this misalignment when the level of stochasticity is high, which is typically the case in diffusion models. We consider that this is an interesting result as well, but unfortunately we don’t have space to discuss this in detail in the main text.
> We believe it is crucial to distinguish between systems that are:
> * **Truly robust**: resistant to adversarial examples regardless of how attacks are optimized.
> * **Non-robust but hard-to-attack**: vulnerable in principle, but attacks fail due to optimization difficulty (e.g., due to noise).
> * **Non-robust and easy-to-attack**: both vulnerable and easy to break.
>
> Our theory in this paper identifies that robustness derived from compression corresponds to the first category. In contrast, robustness from randomness alone places the system in the second category. The practical implication is critical: systems in the second category may **appear robust under current evaluations but still exhibit dangerous failure modes** in real-world deployments.
>
> For instance, imagine two self-driving cars. The first relies on a compression-based purification system that consistently maps perturbed inputs back to their anchor points. The second relies solely on large internal randomness to thwart gradient-based attacks. While both may resist adversarial attacks during benchmarking, only the first can be expected to behave reliably in natural environments. The second may still encounter adversarial failures triggered unintentionally—resulting in sporadic, unexplained accidents.
>
> ---
>
> Thank you for your thoughtful review. We are happy to address any further questions and would appreciate it if you could share your understanding with the other reviewers during the discussion period.

---

### Official Review · Reviewer_x1cW · 2025-10-29

**Soundness:** 3
**Presentation:** 3
**Contribution:** 3
**Rating:** 4
**Confidence:** 3

**Summary:**

This paper challenges conventional views on diffusion-based purification (DBP) robustness by proposing a compression-based perspective. The authors show that diffusion models compress image space when randomness is fixed, and introduce the compression rate metric to quantify this effect. They establish a sigmoidal relationship between CR and robustness, providing theoretical insights that compression primarily affects off-manifold perturbations through convergent score fields.

**Strengths:**

1. The paper is clearly written. It identifies flaws in existing explanations, proposes a novel compression-based framework, and substantiates it with both theoretical analysis and empirical evidence. The visualizations effectively assists in conveying the key ideas.

2. The paper provides a fresh perspective in shifting the focus from "denoising" to "space compression" as the mechanism underlying DBP robustness. The introduction of compression rate linkes purification behavior to adversarial robustness, and the distinction between on-manifold and off-manifold perturbation behavior provides insights.

3. The proposed compression rate offers potential value—it can be adopted as a tool to evaluate adversarial robustness of purification models, facilitating more efficient model robustness assessment.

**Weaknesses:**

1. The experimental validation is conducted on a relatively narrow set of models and datasets, this makes it difficult to assess whether the compression-robustness relationship holds universally or only under specific conditions.

2. The paper lacks implementation details for computing the compression rate metric, also no ablation studies are presented to demonstrate CR's stability with respect to hyperparameters. Without these analyses, it is unclear whether CR is a robust and reliable metric in practice or sensitive to implementation choices.

**Questions:**

How are on-manifold and off-manifold perturbations mathematically connected to CR in the derivation? What specific property distinguishes them such that compression selectively affects off-manifold perturbations while preserving on-manifold components?

---

> ### Author Response · Authors · 2025-11-28
>
> We thank the reviewer for providing constructive feedback and in general positive reviews. Please find our responses to the weaknesses and questions below.
>
> # Weakness 1: Experimental validation on broader dataset and sampling methods
> This is a valid concern. To address this, we implemented two continuous sampling methods, VPSDE [1] and DPM [2], and tested on the ImageNet dataset. Our study initially excludes continuous diffusion models due to the concern that, in continuous dynamical systems, the stochasticity introduced by numerical solvers (e.g., adaptive stepsizes) can induce gradient masking [3]. This issue does not arise for VPSDE or DPM samplers when they are implemented with fixed stepsizes. The resulting compression rates are shown below.
>
> ### Table: Compression rates and robustness predictions on ImageNet
> | **Method** | **Clean Acc.** | **Expected CR** |  Pred Robust. (Clean) | Pred Robust. (Anchor) |
> |-----------|----------------|-----------------|-----------------|--------|
> | DDPM      | 73.6%          | 0.147 $\pm$ 0.070 |  46.1\% | 42.2\% |
> | VPSDE     | 75.4%          | 0.165 $\pm$ 0.059 | 40.1\% | 38.0\% |
> | DPM-20    | 59.8%          |0.337 $\pm$ 0.015 | 1.0\%   | 0.5%    |
>
> Again, we observed a substantial compression of the image space toward anchor points across all diffusion models. Based on the expected compression rates, our theory predicts that DDPM should exhibit slightly higher robustness than VPSDE, while DPM-20 should be significantly worse.
>
> We additionally examined the distribution of the classifier’s minimum perturbations (ResNet-50). The distribution remains a zero-inflated Gaussian—consistent with our observations on CIFAR-10. Consequently, the survival function (1–CDF) again exhibits a sigmoidal form, enabling us to predict the robustness of DDPM, VPSDE, and DPM-20 on ImageNet directly from their expected compression rates. Due to limited computational resources, we were unable to run full PGD attacks on ImageNet. Nevertheless, our released code will include full support for PGD attacks on ImageNet for future evaluation.
>
> For completeness, we list below the corresponding prediction results on CIFAR-10:
>
> ### Table: Compression rates and robustness predictions on CIFAR-10
> | **Method** | **Clean Acc.** | **Expected CR** | **Pred. Robust. (Clean)** | **Pred. Robust. (Anchor)** | **PGD Robust.** |
> |--------------|------------------|-----------------|----------------------------|----------------------------------|----------------------|
> | DDPM      | 85.5%          | 0.231 $\pm$ 0.046 | 27.8%                     | 26.0%                      | 23.7%            |
> | Reverse   | 87.2%          | 0.195 $\pm$ 0.046 | 45.3%                     | 40.9%                      | 34.9%            |
> | DDIM      | 87.5%          | 0.281 $\pm$ 0.048 | 10.2%                     | 10.6%                      | 10.9%            |
>
> The predictions are quantitatively accurate—especially considering that robustness changes extremely rapidly in the steep region of the sigmoid. For example, for DDPM, the range $0.231 \pm 0.046$ corresponds to a predicted robustness between 10.6% and 48.1% at the anchor point (where robustness is most sensitive to small changes in compression rate). In contrast, robustness varies slowly at very small or very large compression rates.
>
> Overall, these extended experiments show that both components of our framework—the **compression effect** (from the diffusion model) and the **sigmoidal relation** (from the classifier)—hold across simple (CIFAR-10) and complex (ImageNet) datasets, and across discretized (DDPM, Reverse-only DDPM, DDIM) as well as continuous (VPSDE, DPM) diffusion models. We believe these results strengthen the empirical and theoretical contributions of our work, highlighting our work as not only a theoretical explanation of robustness but also a resource-efficient robustness prediction method. We will incorporate those additional prediction results in the revised version.

---

> ### Author Response · Authors · 2025-11-28
>
> # Weakness 2: Reliability analysis and ablation study on the compression rates
>
> Thanks for this suggestion. For clarification, we use uniform noise, with the perturbation scale $\epsilon$ matches the corresponding adversarial perturbation (that is, 8 / 255 for CIFAR-10 and 4 / 255 for ImageNet) to calculate the expected CR and conveniently control the $\ell_\infty$ norm. We will include the details in the revised version.
>
> Given the fact that our CR definition is extremely simple, the only hyperparameter involved is the scale of the random perturbation $\epsilon$. Theoretically, the scale should not matter under our definition: the scale factor in the numerator and denominator cancels out, as long as $\epsilon$ is sufficiently small for the first-order Taylor expansion to remain valid—an assumption central to our analysis in Section 6. Empirically, as stated above, we did not tune $\epsilon$, but instead, used the standard adversarial scales for each dataset in the original submission (thus, even for this parameter, there is no tuning involved).
>
> To further examine the relationship between perturbation magnitude and compression rate, we measured the expected CR under random perturbations scaled from $1/4\times$ to $4\times$ the typical adversarial magnitude. The results are shown below.
>
> ### Table: Compression rates vs. perturbation scale (DDPM, CIFAR-10)
>
> | **Epsilon** | **2 / 255** | **4 / 255** | **8 / 255** | **16 / 255** | **32 / 255** |
> |-------------|-------------|-------------|-------------|--------------|--------------|
> | **Expected CR** | 0.2298 $\pm$ 0.0469 | 0.2302 $\pm$ 0.0461 | 0.2306 $\pm$ 0.0457 | 0.2388 $\pm$ 0.0488 | 0.2676 $\pm$ 0.0568 |
>
> As shown above, for small $\epsilon$ (up to $16/255$), the compression rate remains very stable. Only when $\epsilon$ becomes noticeably large (e.g., $32/255$) does the CR begin to drift, which is fully consistent with our theoretical expectation that the Taylor approximation breaks down at large perturbation scales.
>
> We also verified that the expected CR is stable across different random seeds:
>
> ### Table: Compression rates across random seeds (DDPM, CIFAR-10)
>
> | **Seed** | **0** | **123** | **295** |
> |----------|-------|---------|----------|
> | **Expected CR** | 0.2306 $\pm$ 0.0457 | 0.2309 $\pm$ 0.0481 | 0.2312 $\pm$ 0.0481 |
>
> Overall, the expected CR is a simple yet remarkably stable quantity that reflects the intrinsic compression capability of diffusion models. We will include these additional results in the revised version.
>
>
> # Question: Connection between the compression rate (CR) and on/off-manifold perturbations
> This is an excellent question, and we are happy to provide further clarification. The CR can be computed effectively along the eigendirections of the Jacobian of the score field. This observation motivates our definition of on-manifold directions as those associated with large eigenvalues, and off-manifold directions as those with small eigenvalues. In Gaussian score fields, this matches the intuition from PCA: large-variance directions correspond to on-manifold structure, while small-variance directions correspond to off-manifold noise.
>
> The expected CR captures the average behavior of this eigenspectrum. Because the sum of eigenvalues equals the trace of the Jacobian, the expected CR is directly linked to the divergence of the score field. Under the common low-dimensional data manifold assumption (e.g., [4] estimates intrinsic dimensions of ~30 for CIFAR-100 and ~45 for ImageNet, consistent with our choice of 40 in the simulation of Figure 4), the average is dominated by the many off-manifold directions. Consequently, the expected CR primarily reflects compression along these off-manifold directions.
>
> Importantly, Figure 4 empirically validates this theory: the overall CR initially aligns with the off-manifold CR and only later converges to the on-manifold CR. This behavior matches our theoretical prediction, indicating that our framework is not ad-hoc but is supported by numerical evidence.
>
> ---
> ### References
> [1] Song, Y., Sohl-Dickstein, J., Kingma, D. P., Kumar, A., Ermon, S., & Poole, B. Score-Based Generative Modeling through Stochastic Differential Equations. In International Conference on Learning Representations, 2021.
>
> [2] Lu, C., Zhou, Y., Bao, F., Chen, J., Li, C., & Zhu, J. (2022). Dpm-solver: A fast ode solver for diffusion probabilistic model sampling in around 10 steps. Advances in neural information processing systems, 35, 5775-5787.
>
> [3] Huang, Yifei, Yaodong Yu, Hongyang Zhang, Yi Ma, and Yuan Yao. "Adversarial robustness of stabilized neural ode might be from obfuscated gradients." In Mathematical and Scientific Machine Learning, pp. 497-515. PMLR, 2022.
>
> [4] Pope, P., Zhu, C., Abdelkader, A., Goldblum, M., & Goldstein, T. The Intrinsic Dimension of Images and Its Impact on Learning. In International Conference on Learning Representations (ICLR), 2021.

---

### Official Review · Reviewer_EPK2 · 2025-10-31

**Soundness:** 4
**Presentation:** 4
**Contribution:** 2
**Rating:** 4
**Confidence:** 4

**Summary:**

This paper investigates how diffusion models improve adversarial robustness. The key contributions are: (1) showing that diffusion models surprisingly increase rather than decrease distances to clean samples, with internal randomness dominating the output; (2) proposing a compression framework where robustness follows a lawful sigmoidal relationship with the compression rate (CR) when randomness is fixed; (3) proving theoretically that compression arises from the convergent score field, with CR capturing off-manifold compression while preserving on-manifold perturbations.

**Strengths:**

1. The paper is well-written and logically organized, guiding the reader from intuitive hypotheses to formal derivations and validation. Figures effectively illustrate the core ideas.

2. The introduction of the expected compression rate (CR) as a measurable quantity connecting diffusion dynamics to robustness is elegant and potentially generalizable to other generative purification systems.

**Weaknesses:**

1. In practical scenarios, an attacker cannot fix the random noise in the defender's diffusion model. The ability to defend against adversarial attacks through randomness is precisely one of the major highlights of diffusion-based purification. However, the paper states "we focus on understanding robustness improvements in diffusion models without randomness." The paper's conclusions may not be applicable to diffusion models with randomness, making them far from practical reality. The paper should verify whether the conclusions hold under diffusion models with randomness (with EOT).

2. Figure 4 suggests that diffusion models perform strong compression off-manifold and weak compression on-manifold. However, Figure S4 shows that clean accuracy also decreases rapidly as the compression rate decreases, indicating that on-manifold are also strongly compressed, which contradicts the conclusion in Figure 4. This suggests that high compression rates may merely be a manifestation of overfitting in the diffusion model.

3. Some ideas (e.g., deterministic evaluation and fixed randomness settings) are closely related to prior work such as DW-Box (Liu et al., 2025). The paper could better clarify its novelty and distinction from concurrent findings.

4. **Limited practical guidance:** While the paper identifies compression as the key mechanism, it provides limited actionable insights for designing better purification systems beyond the two criteria mentioned in the discussion (high clean accuracy and strong compression). For instance, how can one directly optimize for compression during training? Are there architectural choices that promote better compression properties?

**Questions:**

Can you provide purified image example with different compression rate? Does the lower compression rate means the worse visual quailty?

---

> ### Author Response · Authors · 2025-11-28
>
> We thank the reviewer for providing thoughtful feedback. Please find our replies below.
>
> # Weakness 1: Discussion on the effect of randomness in adversarial robustness
> We appreciate the reviewer’s question regarding how to properly understand the role of randomness in adversarial robustness. This is indeed an important yet subtle issue, and we believe part of the disagreement may stem from differing views on what constitutes true robustness. Without clarifying this point, the main compression argument of our paper may not be fully appreciated.
>
> Our argument can be summarized as follows: randomness will make a system **hard-to-attack, but not truly robust**. In other words, randomness helps by making the system harder for an adaptive adversary to optimize against, but it does not fundamentally change the model’s behavior under naturally occurring perturbations. We provide a more detailed explanation below and are happy to further discuss this perspective.
>
> Our central argument is that the evaluated empirical robustness of purification systems can be **decomposed into two components**: (1) the part that arises from intrinsic properties of the system when randomness is fixed, and (2) the part that results solely from stochasticity during the attack and defense phases. For diffusion-based purification, we show that the first component can be explained through our proposed compression theory. Importantly, this decomposition applies not only to diffusion models but to all adversarial purification systems involving randomness.
>
> As analyzed in Section 4 and Appendix B, randomness appears to “improve” robustness because it breaks the consistency between the gradients used to generate adversarial examples and the gradients seen during defense. In other words, the attack and defense use **different noise realizations**, making the **attack gradients suboptimal or even meaningless**. This may be interpreted as an artifact of evaluation, not a true indication of robustness.
>
> Importantly, based on our analysis, the effect of different randomness realizations can be interpreted as a form of transfer attack. As shown in Table S4, the robustness without controlling for randomness resembles the behavior observed when adversarial examples are transferred across different randomness configurations. In this setting, the computed adversarial directions are only weakly correlated with the true optimal direction. Moreover, as demonstrated both analytically and empirically in Table S5, Expectation over Transformation (EOT) fails to resolve this misalignment when the level of stochasticity is high, which is typically the case in diffusion models. We consider that this is an interesting result as well, but unfortunately we don’t have space to discuss this in detail in the main text.
>
> We believe it is crucial to distinguish between systems that are:
> * **Truly robust**: resistant to adversarial examples regardless of how attacks are optimized.
> * **Non-robust but hard-to-attack**: vulnerable in principle, but attacks fail due to optimization difficulty (e.g., due to noise).
> * **Non-robust and easy-to-attack**: both vulnerable and easy to break.
> Our theory in this paper identifies that robustness derived from compression corresponds to the first category. In contrast, robustness from randomness alone places the system in the second category. The practical implication is critical: systems in the second category may **appear robust under current evaluations but still exhibit dangerous failure modes** in real-world deployments.
>
> For instance, imagine two self-driving cars. The first relies on a compression-based purification system that consistently maps perturbed inputs back to their anchor points. The second relies solely on large internal randomness to thwart gradient-based attacks. While both may resist adversarial attacks during benchmarking, only the first can be expected to behave reliably in natural environments. The second may still encounter adversarial failures triggered unintentionally—resulting in sporadic, unexplained accidents.

---

> ### Author Response · Authors · 2025-11-28
>
> # Weakness 2: Compression of the on-manifold directions in the high-compression rate regime
>
> For an adversarial purification system to be practical, two conditions must be met: (1) the anchor point must retain reasonably high clean accuracy, and (2) there must be sufficient compression toward the anchor point so that the robust accuracy approaches the anchor point’s clean accuracy. In the high-$T$ regime, although the compression rate (CR) increases, the clean accuracy at the anchor points drops substantially. As the reviewer correctly noted, this makes the high-$T$ regime unsuitable for adversarial purification despite its large CR.
>
> More precisely, in Figure 3a, each point corresponds to its own sigmoidal relation (arising from the distribution of minimum perturbations around its anchor point), with the curve capped by the anchor point’s clean accuracy at the y-intercept. In the regime where adversarial purification remains meaningful—i.e., where the clean accuracy at the anchor points does not significantly deteriorate relative to that of the original samples ($T \leq 150$)—the sigmoidal relation at the anchor points is well approximated by that of the clean samples. We omitted these details in the main submission due to space constraints, but we will clarify them in the revision.
>
> In summary, our theory states that robustness is governed jointly by: (i) the clean accuracy at the anchor point, and (ii) the amount of compression (CR) toward that anchor point. In the large-$T$ regime, although the CR is high, the anchor points’ clean accuracy becomes too low, and therefore no substantial robustness improvement can be achieved.
>
> # Weakness 3: Comparison with DW-Box in evaluating the robustness of diffusion models with fixed randomness
>
> In the second half of Section 4, we proposed evaluating the robustness of diffusion models under fixed randomness. As stated in the main text:
>
> > A concurrent work argued for a similar evaluation protocol (DW-Box (Liu et al., 2025)) under slightly different model settings. Beyond their analysis, our results further suggest that in the presence of randomness, EOT should instead be understood as a transfer attack (Appendix B).
>
> We acknowledge the concurrent development of this protocol and explicitly credit their contribution. However, fixing the randomness only addresses the question of **how** to evaluate the robustness of diffusion models. It does not address the deeper and more fundamental question of **why** diffusion models can improve robustness—even when randomness is fixed. This is precisely the main question that our compression theory addressed, and it is the central focus of our paper, as reflected in the title.
>
> # Weakness 4: Practical guidance of our compression theory
>
> Our theory shifts the focus of understanding robustness in diffusion models from the clean images to the anchor points. This perspective suggests that neither improving clean-image reconstruction nor optimizing the FID score (the primary objective in generative modeling) is directly aligned with improving adversarial robustness in diffusion models.
>
> Based on our theory, we identify two concrete and actionable future directions:
> 1. Finetuning existing diffusion models using a compression loss toward the anchor points, encouraging stronger contraction around robust anchor representations;
> 2. Constructing compression-based purification systems from scratch, with the condition that (i) the anchor points achieves high classification accuracy, (ii) a high CR towards such anchor points $f(x_0)$.
>
> Overall, our theory decouples the traditional goal of image generation from the goal of improving robustness and shows that compression is the primary mechanism for robustness in diffusion models.
>
> # Question: Additional examples of purified states and visual quality
>
> We will include additional examples of purified states under different CR values in the revised version. Across these samples, we did not observe noticeable degradation in visual quality.
>
> For conceptual clarity, we emphasize that our compression rate (CR) measures the **compression of the image space**, which is fundamentally **different from image compression in the sense of reducing information bits**. In fact, we have results showing that standard image compression algorithms such as JPEG do **not** effectively compress the image space (their CR is greater than 1). This highlights the distinction between these two notions of compression: visual quality is more directly related to information bits compression, whereas our CR quantifies geometric compression in representation space. We are happy to incorporate these results if the reviewer believes they would be helpful for improving the paper.

---

### Official Review · Reviewer_rBUd · 2025-10-31

**Soundness:** 3
**Presentation:** 3
**Contribution:** 3
**Rating:** 4
**Confidence:** 3

**Summary:**

This paper investigates how diffusion models improve adversarial robustness through purification. Contrary to the common belief that purification "denoises" adversarial examples closer to clean images, the authors find that diffusion models actually increase the distance to clean samples. They argue that the robustness gain comes from compression of the image space, not from alignment with clean data. The authors propose a compression rate (CR) metric and show a sigmoidal relationship between CR and adversarial robustness under fixed randomness. Theoretical analysis further supports that diffusion models compress off-manifold perturbations while preserving on-manifold structure.

**Strengths:**

1.	Originality: The paper challenges two prevalent intuitive hypotheses regarding diffusion models' adversarial robustness and proposes a novel image space compression framework. The introduced compression rate (CR) and its sigmoidal relationship with robustness are original findings.
2.	Quality: Extensive experiments across datasets, attacks, and sampling methods support the claims.
3.	Clarity: The paper is clearly written and well-structured, with intuitive presentations of data and results that facilitate understanding.
4.	Significance: The work provides a principled understanding of diffusion-based purification mechanisms and offers practical insights for designing more robust models, making it a valuable contribution to the field.

**Weaknesses:**

1.	Insufficient Analysis of Sampling Methods: The paper lacks a thorough comparative analysis of the compression rates and robustness of sampling techniques, such as VPSDE (DDPM++) [1] and DPEDM [2]. A more in-depth evaluation of these methods would strengthen the empirical contributions of the work.
2.	Lack of Semantic Similarity Analysis: While the paper discusses the increase in distance measures and emphasizes perceptual similarity using SSIM, it does not investigate semantic similarity. It is recommended to include CLIP-based similarity metrics to assess whether the outputs of diffusion models are semantically closer to the clean images.
3.	Unverified Manifold Hypothesis: The theoretical analysis is grounded in the assumption of a low-dimensional manifold structure, yet this hypothesis is not empirically validated using real image data.
4.	Incomplete Visualization: The compelling relationship in Figure 3 could be further strengthened by incorporating all data points from Table 3. Including more DDIM results would also help better illustrate the generalizability of the sigmoidal trend across different model configurations.


[1] Yang Song, Jascha Sohl-Dickstein, Diederik P Kingma, Abhishek Kumar, Stefano Ermon, and Ben Poole. Score-based generative modeling through stochastic differential equations. In International Conference on Learning Representations.
[2] Tero Karras, Miika Aittala, Timo Aila, and Samuli Laine. Elucidating the design space of diffusion-based generative models. Advances in Neural Information Processing Systems, 35:26565–26577, 2022.

**Questions:**

1.	How does the compression rate (CR) metric behave beyond the infinitesimal perturbation regime? Does the sigmoidal relationship between CR and adversarial robustness remain valid for larger perturbation magnitudes?

**Details Of Ethics Concerns:**

None.

---

> ### Author Response · Authors · 2025-11-28
>
> We thank the reviewer for the constructive feedback. Please find our responses to the identified weaknesses below.
>
>
> # Weakness 1: Additional continuous sampling methods (VPSDE, DPEDM)
>
> Thank you for this suggestion. Our study initially focused on discretized diffusion models because, in continuous dynamical systems, the stochasticity introduced by numerical solvers (e.g., adaptive stepsizes) can induce gradient masking [1]. This issue does not arise for VPSDE or DPM [2] samplers when they are implemented with fixed stepsizes.
>
> To address the reviewer’s concern, we implemented VPSDE and DPM sampling on the ImageNet dataset (EDM only provides checkpoints for conditional ImageNet generation, which is incompatible with adversarial purification). The resulting compression rates are shown below.
>
> ### Table: Compression rates and robustness predictions on ImageNet
> | **Method** | **Clean Acc.** | **Expected CR** |  **Pred Robust. (Clean)** | **Pred Robust. (Anchor)** |
> |--------------|-----------------|---------------------|-----------------|---------------|
> | DDPM      | 73.6%          | 0.147 $\pm$ 0.070 |  46.1\% | 42.2\% |
> | VPSDE     | 75.4%          | 0.165 $\pm$ 0.059 | 40.1\% | 38.0\% |
> | DPM-20    | 59.8%          | 0.337 $\pm$ 0.015 | 1.0\%   | 0.5%    |
>
> Again, we observed a substantial compression of the image space toward anchor points across all diffusion models we tested above. Based on the expected compression rates, our theory predicts that DDPM should exhibit slightly higher robustness than VPSDE, while DPM-20 should be significantly worse.
>
> We additionally examined the distribution of the classifier’s minimum perturbations (ResNet-50). The distribution remains a zero-inflated Gaussian, consistent with our observations on CIFAR-10. Consequently, the survival function (1–CDF) again exhibits a sigmoidal form, enabling us to predict the robustness of DDPM, VPSDE, and DPM-20 on ImageNet directly from their expected compression rates. Due to the limited computational resources we have, we were unable to run full PGD attacks on ImageNet. Nevertheless, our released code will include full support for PGD attacks on ImageNet for future evaluation.
>
> For completeness, we list below the corresponding prediction results on CIFAR-10:
>
> ### Table: Compression rates and robustness predictions on CIFAR-10
> | **Method** | **Clean Acc.** | **Expected CR** | **Pred. Robust. (Clean)** | **Pred. Robust. (Anchor)** | **PGD Robust.** |
> |-----------|----------------|-----------------|----------------------------|-----------------------------|------------------|
> | DDPM      | 85.5%          | 0.231 $\pm$ 0.046 | 27.8%                     | 26.0%                      | 23.7%            |
> | Reverse   | 87.2%          | 0.195 $\pm$ 0.046 | 45.3%                     | 40.9%                      | 34.9%            |
> | DDIM      | 87.5%          | 0.281 $\pm$ 0.048 | 10.2%                     | 10.6%                      | 10.9%            |
>
>
> The predictions are quantitatively accurate—especially considering that robustness changes extremely rapidly in the steep region of the sigmoid function. For example, for DDPM, the range 0.231 $\pm$ 0.046 corresponds to a predicted robustness between 10.6% and 48.1% at the anchor point (where robustness is most sensitive to small changes in compression rate). In contrast, robustness varies slowly at very small or very large compression rates.
>
> Overall, these extended experiments further support our main results. They show that both components of our framework—the **compression effect** (from the diffusion model) and the **sigmoidal relation** (from the classifier)—hold across simple (CIFAR-10) and complex (ImageNet) datasets, and across discretized (DDPM, Reverse-only DDPM, DDIM) as well as continuous (VPSDE, DPM) diffusion models. We believe these results strengthen the empirical and theoretical contributions of our work, highlighting our work as not only a theoretical explanation of robustness but also a resource-efficient robustness prediction method. We will incorporate those additional prediction results in the revised version.

---

> ### Author Response · Authors · 2025-11-28
>
> # Weakness 2: Semantic similarity analysis (CLIP)
>
> This is an interesting suggestion. We further measured the cosine distance between the adversarial samples and their corresponding clean samples in the CLIP representation space, both before and after purification. As shown below, we observed a reduction in distance for smaller timesteps $t$, whereas for larger $t$ the distance increases.
>
> ### Table: CLIP distance reduction after purification
> | **Method**       | **t=10** | **t=20** | **t=50** | **t=100** | **t=150** |
> |------------------|----------|----------|----------|-----------|-----------|
> | DDPM         | 0.0285   | 0.0249   | 0.0082   | -0.0162   | -0.0330   |
> | Reverse          | 0.0456   | 0.0359   | 0.0100   | -0.0135   | -0.0272   |
> | DDIM    | 0.0327   | 0.0333   | 0.0195   | 0.0009    | -0.0131   |
>
> Similarly, if we treat the purified CLIP distance to the clean samples as a robustness indicator, the results are shown below.
>
> ### Table: CLIP cosine distance after purification
> | **Method**       | **t=10** | **t=20** | **t=50** | **t=100** | **t=150** |
> |------------------|----------|----------|----------|-----------|-----------|
> | DDPM         | 0.0793   | 0.0778   | 0.0815   | 0.0939    | 0.1056    |
> | Reverse          | 0.0624   | 0.0619   | 0.0703   | 0.0840    | 0.0948    |
> | DDIM    | 0.0758   | 0.0740   | 0.0798   | 0.0905    | 0.1002    |
>
> Overall, similar to the FID results, neither FID nor CLIP semantic distances exhibit a consistent monotonic relationship with robustness, indicating that they are not reliable robustness indicators. This contrasts with the strong and consistent correlation observed with our proposed expected compression rate, further supporting our claim that robustness arises from a global compression of the image space rather than proximity to clean samples in FID or semantic space. We will incorporate these results to the revised version of the paper.
>
>
> # Weakness 3: Justification of the low-dimensional manifold hypothesis for natural stimuli
>
> Thank you for this question and we are happy to clarify. Although we referred to it as a hypothesis, the low-dimensional manifold assumption is relatively well-established in computational vision.
>
> Intuitively, consider CIFAR-10, which is embedded in a 32×32×3 = 3072-dimensional space. Random sampling from this 3072-dimensional pixel space would almost never produce a structured CIFAR-10 image, indicating that the intrinsic dimension of the CIFAR-10 manifold is much lower than the embedding dimension.
>
> This observation has also been studied more formally. For example, [2] estimated the intrinsic dimension of CIFAR-100 to be around 30 and ImageNet to be around 45, both much lower than their corresponding embedding dimensions. In our numerical simulations of the theory (Figure 4), we assume an intrinsic dimension of 40 (as listed in Table S1), which is consistent with these previous estimations. We will add the reference and make this point more explicit in the revised version.
>
>
> # Weakness 4: Additional DDIM results in Figure 3
>
> Thank you for this suggestion. We extended the DDIM experiments to cover the full range of timesteps (previously, we only reported the point at $t = 100$). The results are shown below:
>
> | **Method** | **Metric**       | **t=10** | **t=20** | **t=50** | **t=100** | **t=150** |
> |------------|------------------|----------|----------|----------|-----------|-----------|
> | **DDIM**   | Expected CR       | 0.641| 0.499    |   0.364  | 0.281 | 0.241       |
> |            | PGD Robustness   | 0.001    | 0.004    | 0.022    | 0.109     | 0.156     |
>
> Running each PGD evaluation requires roughly 20 hours, while computing the Expected CR takes only ~20 minutes. In the updated version of the paper, we therefore restrict the timestep range to $t \le 150$.
>
> The deeper reason is that as $t$ increases, the *clean accuracy at the anchor points* consistently decreases. Once the clean accuracy becomes too low, the sigmoidal curve at the anchor point can no longer be well-approximated by the sigmoidal curve fitted on the clean samples—violating a key prerequisite of our method (i.e., anchor points must have reasonably high clean accuracy). This behavior is not unique to DDIM: we already observed the same issue for Reverse-only DDPM at $t=300$ in the original submission, but did not elaborate due to space constraints. For further discussion, please refer to our response to Weakness 1 from reviewer **xbFW**.
>
> Nonetheless, *within the practically relevant robustness range* ($t \le 150$, where anchor-point accuracy remains close to that of clean samples), the additional DDIM points still lie on the **same robustness–compression curve** as the datapoints from all other methods. This further supports and validates our theoretical framework. We will integrate these additional results into the revised **Figure 3**.

---

> ### Author Response · Authors · 2025-11-28
>
> # Question: Compression rates at large-magnitude perturbations
>
> This is an interesting question. Theoretically, the scale of the perturbation $\epsilon$ should not matter under our definition: the scale factor in the numerator and denominator cancels out, as long as $\epsilon$ is sufficiently small for the first-order Taylor expansion to remain valid—an assumption central to our analysis in Section 6.
>
> Empirically, the magnitude of  $\epsilon$ is set according to standard adversarial scales for each dataset, i.e., $\epsilon = 8/255$ for CIFAR-10 and $\epsilon = 4/255$ for ImageNet. To further examine the relationship between perturbation magnitude and compression rate, we measured the expected compression rate (CR) under random perturbations scaled from $1/4\times$ to $4\times$ the typical adversarial magnitude. The results are shown below.
>
> ### Table: Compression rates vs. perturbation scale (DDPM, CIFAR-10)
>
> | **Epsilon** | **2 / 255** | **4 / 255** | **8 / 255** | **16 / 255** | **32 / 255** |
> |-------------|-------------|-------------|-------------|--------------|--------------|
> | **Expected CR** | 0.2298 $\pm$ 0.0469 | 0.2302 $\pm$ 0.0461 | 0.2306 $\pm$ 0.0457 | 0.2388 $\pm$ 0.0488 | 0.2676 $\pm$ 0.0568 |
>
> As shown above, for small $\epsilon$ (up to $16/255$), the compression rate remains very stable. Only when $\epsilon$ becomes noticeably large (e.g., $32/255$) does the compression rate (CR) begin to drift, which is fully consistent with our theoretical expectation that the Taylor approximation breaks down at large perturbation scales.
>
> We also verified that the expected compression rate (CR) is stable across different random seeds:
>
> ### Table: Compression rates across random seeds (DDPM, CIFAR-10)
>
> | **Seed** | **0** | **123** | **295** |
> |----------|-------|---------|----------|
> | **Expected CR** | 0.2306 $\pm$ 0.0457 | 0.2309 $\pm$ 0.0481 | 0.2312 $\pm$ 0.0481 |
>
> Overall, the expected CR is a simple yet remarkably stable quantity that reflects the intrinsic compression capability of diffusion models. We will include these additional results in the revised version.
>
> ---
> ### References
> [1] Huang, Yifei, Yaodong Yu, Hongyang Zhang, Yi Ma, and Yuan Yao. "Adversarial robustness of stabilized neural ode might be from obfuscated gradients." In Mathematical and Scientific Machine Learning, pp. 497-515. PMLR, 2022.
>
> [2] Lu, C., Zhou, Y., Bao, F., Chen, J., Li, C., & Zhu, J. (2022). Dpm-solver: A fast ode solver for diffusion probabilistic model sampling in around 10 steps. Advances in neural information processing systems, 35, 5775-5787.
>
> [3] Pope, P., Zhu, C., Abdelkader, A., Goldblum, M., & Goldstein, T. The Intrinsic Dimension of Images and Its Impact on Learning. In International Conference on Learning Representations (ICLR), 2021.

---

### Author Response · Authors · 2025-12-04
**Summary of Rebuttal and Revisions for the Area Chair**

Dear Area Chair,

For your convenience, we summarize below the main points of our rebuttal and the corresponding revisions to the draft:

* **Experiments on ImageNet with continuous sampling methods (VPSDE, DPM-20)** (Reviewers rBUd, x1cW; Sec. 5.3, Tables 3, 4)
  We extend our analysis to the more complex ImageNet dataset and to continuous-time samplers. The identified compression mechanism consistently holds, addressing the concern that compression is specific to discrete diffusion models on CIFAR-10.

* **Reliability analysis of compression rates** (Reviewers rBUd, x1cW; Sec. 5.3, Tables 5, 6)
   We show that the proposed expected compression rate is stable across a wide range of perturbation magnitudes and random seeds.

* **Distribution of minimum perturbations at anchor points** (Reviewer xbFW; Appendix B)
  We demonstrate that the distribution of minimum perturbations at anchor points closely resembles that at clean samples (Fig. S1), which supports more accurate robustness estimation (Tables 3, 4).

* **Examples of adversarial and purified images** (Reviewers EPK2, x1cW; Appendix H)
We clarify that our compression rate concerns the geometric compression of the image space, and should not be conflated with information-theoretic compression affecting visual quality.

* **Conceptual clarifications**
   We explain how randomness should be interpreted in robustness evaluation (Reviewers EPK2, xbFW). We add justification of the low-dimensional manifold hypothesis with appropriate citation (Reviewer rBUd).

* **Additional experiments**
   We further show that FID and semantic distances fail to account for robustness improvements with experiments (Fig. S5 and Appendix E3). Additional DDIM results (Table 2) follow the same sigmoidal relation (Fig. 3).

Overall, we believe that the reviewers’ questions and concerns are effectively addressed by our rebuttal and the revised manuscript. We appreciate your review and consideration.

---

### Meta-Review · Area_Chair_82eK · 2025-12-28

**Summary:**

This paper proposes a compression-based explanation for adversarial robustness in diffusion-based purification and is generally recognized by reviewers and the AC as original, clearly written, and empirically thorough. The evaluation remained borderline because key claims are not yet fully supported, particularly regarding the practical relevance of results under fixed randomness, the causal role of compression, and the rigor of the manifold-based theoretical analysis. Given the extensive and rigorous existing literature on manifold-based analysis of diffusion models, the current theoretical depth remains limited. Overall, the rebuttal meaningfully strengthens the paper, particularly on the empirical side, and clarifies several important points, but further work is still needed to fully address the central concerns underlying the borderline evaluation.

**Reviewer Concerns:**

The rebuttal addressed several empirical and clarification-related concerns by adding experiments with additional samplers and extended DDIM results, providing CLIP-based semantic analyses alongside FID, examining the stability of the compression rate, and clarifying notations and presentation.

However, several issues remain only partially resolved. The practical relevance of conclusions drawn under fixed randomness for inherently stochastic diffusion defenses is still unclear, and the proposed decomposition between intrinsic robustness and randomness-induced “hard-to-attack” effects remains largely conceptual rather than rigorously established. On the theoretical side, the analysis relies on a low-dimensional manifold assumption without clearly specifying which manifold is being modeled or rigorously connecting score Jacobian spectra to on- versus off-manifold directions. In addition, the observed compression–robustness relationship remains partially confounded by anchor-point accuracy degradation in high-compression regimes.

Moreover, in light of the substantial existing literature on manifold analysis in diffusion models (e.g., [R1–R3]), the Jacobian-based interpretation of on- versus off-manifold behavior appears underdeveloped and would benefit from greater rigor, specificity, and conceptual depth.

[R1] He, Yutong, et al. Manifold Preserving Guided Diffusion. ICLR 2024.

[R2] Chung, Hyungjin, et al. Improving diffusion models for inverse problems using manifold constraints. NeurIPS 2022.

[R3] Stanczuk, Jan Pawel, et al. Diffusion models encode the intrinsic dimension of data manifolds. ICML 2024.

**Reviewer Scores:**

There are four reviewers, with scores of 4, 4, 4, and 6, respectively. Based on the rebuttal, the scores would likely remain largely unchanged. The authors made a constructive effort to address many empirical and clarification-related concerns, strengthening the experimental support and improving clarity. At the same time, several foundational issues, especially the practical relevance under stochastic settings and the rigor of the theoretical interpretation, were only partially resolved and would likely limit upward score revisions.

---

### Decision · Program_Chairs · 2026-01-26

Reject